# Proximal Operators of Non Convex SLOPE

## Abstract

This work studies the problem of sparse structured generalized linear models with sorted nonsmooth penalties, which are known to induce an automatic grouping of the features without a priori knowledge. Generalizing the Sorted L1 Penalty (SLOPE), we introduce a family of *nonconvex sorted penalties* which not only promote clustering of variables, but are less biased than their popular convex counterpart. For sorted weakly convex penalties (e.g. sorted MCP and SCAD), we provide an algorithm that exactly and efficiently computes their proximal operator. Moreover, we show that a slight modification of this algorithm turns out to be remarkably efficient to tackle the computation of the proximal operator of sorted $\ell_q$ with $q \in \,]0, 1[$, which is not weakly convex and whose prox yields a challenging combinatorial problem. We demonstrate the interest of using such penalties on several experiments.

## 1 Introduction

Sparsity and structure are two desirable characteristics for a model as they improve its interpretability and decrease its complexity. Sparsity is most usually enforced through a penalty term favoring variable selection, i.e. solutions that use only a subset of features. Such sparsity-promoting penalties share the common property to be nonsmooth at zero. While being widely used, the convex $\ell_1$ norm (the Lasso, Tibshirani, 1996) has the drawback of shrinking all nonzero coefficients towards 0. In the sparse recovery literature, this undesirable property is commonly called *bias* . It can be mitigated by using nonconvex penalties (Selesnick & Bayram, 2014). These include smoothly clipped absolute deviation (SCAD, Fan & Li, 2001), minimax concave penalty (MCP, Zhang et al., 2013) or the log-sum penalty (Candes et al., 2008). Another very popular class of nonconvex penalties are the $\ell_q$ regularizers (Grasmair et al., 2008), with $q \in \,]0, 1[$. The latter are not only nonconvex and nonsmooth (like MCP and SCAD) but also non-Lipschitz. We refer to Appendix A for an overview of popular nonconvex penalties.

Beyond sparsity, one may be interested in clustering features, i.e. in assigning equal values to coefficients of correlated features. In practice, such groups should be recovered *without prior knowledge about the clusters*. The Lasso is unfit for this purpose as it tends to only select one relevant feature from a group of correlated features. The Fused Lasso (Tibshirani et al., 2005) is a remedy, but only works for features whose ordering is meaningful, as groups can only be composed of consecutive features. Elastic-Net regression (Zou & Hastie, 2003), which combines $\ell_1$ and squared $\ell_2$ regularization, encourages a grouping effect, but does not enforce exact clustering. In contrast, the OSCAR model (Bondell & Reich, 2008) combines a $\ell_1$ and a pairwise $\ell_\infty$ penalty to simultaneously promote sparsity and equality of some coefficients, i.e. clustering. OSCAR has been shown to be a special case of the sorted $\ell_1$ norm regularization (Bogdan et al., 2013; Zeng & Figueiredo, 2014). As our present work heavily builds upon this penalty, we recall its definition below.

**Definition 1.1** (SLOPE (Bogdan et al., 2013), OWL (Zeng & Figueiredo, 2014))**.** *The sorted $\ell_1$ penalty, denoted $\Psi_{\mathrm{SLOPE}}$, writes, for $\mathbf{x} \in \mathbb{R}^p$:*

$$\Psi_{\mathrm{SLOPE}}(\mathbf{x}) = \sum_{i=1}^{p} \lambda_i |x_{(i)}|, \tag{1}$$

*where $x_{(i)}$ is the i-th component of $\mathbf{x}$ sorted by non-decreasing magnitude, i.e. such that $|x_{(1)}| \geq \cdots \geq |x_{(p)}|$, and where $\lambda_1 \geq \cdots \geq \lambda_p \geq 0$ are chosen regularization parameters.*

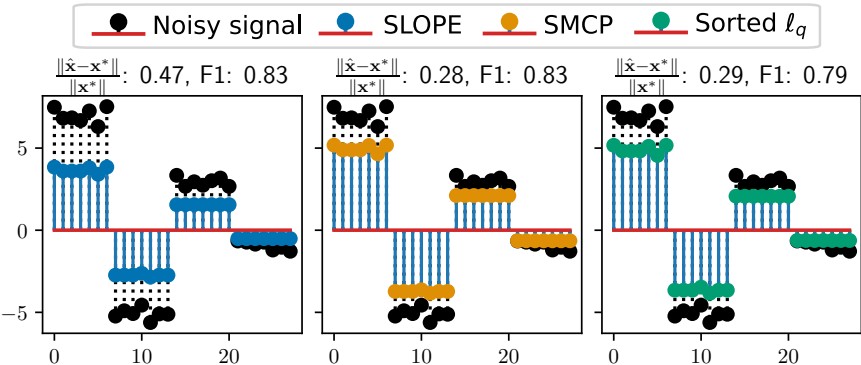

Figure 1: Sorted penalties for denoising: $\hat{\mathbf{x}}$ is the proximal operator of the penalty applied to a noisy version of the ground truth $\mathbf{x}^*$. With equivalent cluster recovery performance (same F1 score), nonconvex sorted penalties ~~mitigate the bias~~ give amplitudes closer to the ground truth compared to convex SLOPE. Note that the clusters are contiguous only for visualization purposes: shuffling the noisy signal does not affect the results. Details on the setup and complete results are in Section 5.1.

The main feature of SLOPE is its ability to exactly cluster coefficients (Figueiredo & Nowak, 2014; Schneider & Tardivel, 2022). Solving many issues of the Lasso, e.g. high False Discovery Rate (Su et al., 2017), SLOPE has attracted much attention in the last years. It has found many applications both in compressed sensing (El Gueddari et al., 2019), in finance (Kremer et al., 2020) or for neural network compression (Zhang et al., 2018). One line of research has focused on its statistical properties: minimax rates (Su & Candès; Bellec et al., 2018), optimal design of the regularization sequence (Hu & Lu, 2019; Kos & Bogdan, 2020), pattern recovery guarantees (Bogdan et al., 2022). Dedicated numerical algorithms were also developed, comprising screening rules (Larsson et al., 2020; Elvira & Herzet, 2023), full path computation à la LARS (Dupuis & Tardivel, 2024) or hybrid coordinate descent methods (Larsson et al., 2023).

Although it mitigates some of the biases of the Lasso, SLOPE is still convex and thus penalizes large coefficients too much. It is thus tempting to use nonconvex variants of SLOPE, replacing the absolute value in Equation (1) by a nonconvex sparse penalty. However, one of the reasons behind the success of SLOPE is its practical usability through the availability of its proximal operator, which can be computed exactly (Zeng & Figueiredo, 2014; Dupuis & Tardivel, 2022). It comes down to solving isotonic regression, which can be done efficiently using the Pool Adjacent Violators (PAV) algorithm (Best & Chakravarti, 1990, recalled in Algorithm 2).

For nonconvex sorted penalties, results are scarcer. A sorted version of MCP was proposed by Feng & Zhang (2019), who studied its statistical properties. The authors also proposed a "majorization-minimization" approach to deal with some sorted nonconvex penalties, using a linear convex approximation of the regularization term. The authors limited their investigation to sorted SCAD and sorted MCP.

In this paper, we consider a large array of nonconvex sorted penalties (sorted versions of SCAD, MCP, Log-Sum, $\ell_q$), that leverage both advantages of nonconvexity and sorting, i.e. unbiasedness and automatic features clustering. Their interest is illustrated on Figure 1, where it is visible that, with equal cluster recovery performance, they ~~decrease the bias of the solution, leading to lower estimation error than SLOPE~~ lead to lower estimation error than SLOPE. To exploit them efficiently, we propose specific algorithms to compute their proximal operators. This allows for their use in combination with virtually any datafit, be it for regression or classification.

**Contribution.** Our contribution is two-fold:

- In Section 3, we show that the proximal operator of **sorted weakly convex penalties** (including sorted MCP and sorted SCAD) can be directly computed: we generalize the PAV algorithm to exactly compute such proximal operators.

- In Section 4, we consider another class of nonconvex penalties, the **sorted $\ell_q$ penalties for** $q \in ]0, 1[$, which, as far as we know, have not been studied yet. It presents an additional level of complexity due to the underlying scalar penalty $\ell_q$ not being Lipschitz continuous. We characterize the local minimizers of the prox minimization problem and provide a PAV-like algorithm which we prove to converge to a subset of them including global minimizer. Moreover, experiments tend to indicate that the proposed PAV-like algorithm may not reach the global solution only for very specific settings and we were not able to build such counter-examples.

**Notation.** Lowercase bold letters such as $\mathbf{x}$ denote vectors. We denote by $x_i$ the $i$-th component of $\mathbf{x}$. For a vector $\mathbf{x}$, we denote by $\mathbf{x}_\downarrow$ the vector obtained by sorting its components by non-increasing magnitude, and $x_{(i)}$ is the $i$-th component of $\mathbf{x}_\downarrow$: $|x_{(1)}| \geq \cdots \geq |x_{(p)}|$. We denote $|\mathbf{x}|$ the element-wise absolute value of a vector $\mathbf{x}$. Note that $|\mathbf{x}|_\downarrow = |\mathbf{x}_\downarrow|$. We denote by $\mathcal{K}_p$ the non-increasing monotone cone: $\mathcal{K}_p := \{\mathbf{x} \in \mathbb{R}^p : x_1 \geq \cdots \geq x_p\}$. In addition, $\mathcal{K}_p^+$ denotes the non-negative non-increasing monotone cone: $\mathcal{K}_p^+ := \mathcal{K}_p \cap \mathbb{R}_+^p$. The sign function simply denoted as sign acts element-wise on some vector $\mathbf{x} \in \mathbb{R}^p$, i.e. $\text{sign}(\mathbf{x}) \in \{-1, 0, 1\}^p$. The positive part is $(\cdot)_+$ and also acts element-wise on vectors. The Hadamard product (i.e. element-wise product) is denoted as $\odot$. For a set of indices $B \subset [\![1, p]\!]$, its cardinality is $|B|$; for some vector $\mathbf{y} \in \mathbb{R}^p$, we define $\bar{y}_B$ as its average value on the block $B$: $\bar{y}_B = \frac{1}{|B|} \sum_{i \in B} y_i$ .

## 2 Background on proximal operator for sorted penalties

In this section, we revisit key properties of sorted penalties, focusing on how the proximal calculation problem simplifies to an isotonic problem. We end this section by recalling the Pool Adjacent Violators (PAV) algorithm, which is the standard method for solving such isotonic problems.

We consider composite penalized problems that write as:

$$\underset{\mathbf{x} \in \mathbb{R}^p}{\arg\min} \, f(\mathbf{x}) + \Psi(\mathbf{x}) \,, \tag{2}$$

where $f$ is an $L-$smooth convex datafit and the function $\Psi$ is the chosen penalty.

Our focus is on $\Psi$, which we take to be a sorted penalty built from a positive scalar penalty $\psi : \mathbb{R} \times \mathbb{R}_+ \to \mathbb{R}_+$, differentiable with respect to its first variable on $\mathbb{R}_+$. To construct $\Psi$, we also fix a non-increasing sequence of regularization parameters $(\lambda_i)_{i \in [\![1, p]\!]}$ such that $\lambda_1 \geq \cdots \geq \lambda_p \geq 0$ (with at least one strict inequality). Then, the sorted penalty writes:

$$\Psi : \mathbf{x} \mapsto \sum_{i=1}^{p} \psi(|x_{(i)}|; \lambda_i) \,. \tag{3}$$

The choices we use[1] for $\psi$ are nonsmooth at 0, hence so is $\Psi$. The most common algorithm to solve Equation (2) is then proximal gradient descent (also referred to as forward-backward splitting, Combettes & Wajs, 2005), which iterates

$$\mathbf{x}^{k+1} = \text{prox}_{\eta \Psi} \left( \mathbf{x}^k - \eta \nabla f(\mathbf{x}^k) \right) \,, \tag{4}$$

where $\eta$ is the stepsize; refinements such as FISTA (Beck & Teboulle, 2009) improve the convergence speed. All these state-of-the-art algorithms rely on the proximal operator of $\Psi$: its computation is mandatory in order to use sorted penalties in practice. The rest of the paper is thus devoted to the computation of $\text{prox}_{\eta \Psi}$, for $\Psi$ of the form in Equation (3).

### 2.1 Basic properties

First, we present general properties of proximal operators for sorted penalties. We recall here, in a more general form, results already known in the case of SLOPE (Definition 1.1). Proximal operators were first studied by Moreau (1965) in the case of proper, lower semi-continuous (l.s.c.) and convex functions. Because most functions under scrutiny in this work are nonconvex, we will introduce here a more general definition (referred to as *proximal mapping*) following the framework of Beck (2017, Chapter 6).

---

[1] MCP, SCAD, $\ell_q$, log, etc.

**Definition 2.1** (Proximal mapping)**.** *The proximal mapping of a function $\Psi : \mathbb{R}^p \rightarrow (-\infty, +\infty]$ is, for any* $\mathbf{x} \in \mathbb{R}^p$:

$$\mathrm{prox}_{\Psi}(\mathbf{x}) = \underset{\mathbf{y} \in \mathbb{R}^p}{\arg\min} \left\{ \Psi(\mathbf{y}) + \frac{1}{2} \|\mathbf{y} - \mathbf{x}\|^2 \right\} .$$

**Proposition 2.2** (Non-emptiness of the prox under closedness and coerciveness, Beck, 2017, Thm. 6.4)**.** *If* $\Psi$ *is proper, l.s.c. and* $\Psi + \frac{1}{2} \| \cdot -\mathbf{x} \|^2$ *is coercive for all* $\mathbf{x} \in \mathbb{R}^p$, *then* $\mathrm{prox}_{\Psi}(\mathbf{x})$ *is nonempty for all* $\mathbf{x} \in \mathbb{R}^p$.

In our setup, since $\Psi$ is bounded below by 0 and thus $\eta\Psi + \frac{1}{2}\| \cdot -\mathbf{x}\|^2$ is coercive, $\mathrm{prox}_{\eta\Psi}$ is nonempty for any stepsize $\eta > 0$. As the next proposition shows, to compute the proximal operator of a sorted penalty, it is enough to be able to compute it on non-negative and non-increasing vectors.

**Proposition 2.3.** *For any* $\mathbf{y} \in \mathbb{R}^p$, *we can recover* $\mathrm{prox}_{\eta\Psi}(\mathbf{y})$ *from* $\mathrm{prox}_{\eta\Psi}(|\mathbf{y}|_{\downarrow})$ *by:*

$$\mathrm{prox}_{\eta\Psi}(\mathbf{y}) = \mathrm{sign}(\mathbf{y}) \odot \mathbf{P}_{|\mathbf{y}|}^{\top} \, \mathrm{prox}_{\eta\Psi}(|\mathbf{y}|_{\downarrow}) ,$$

*where* $\mathbf{P}_{|\mathbf{y}|}$ *denotes any permutation matrix that sorts* $|\mathbf{y}|$ *in non-increasing order:* $\mathbf{P}_{|\mathbf{y}|}(|\mathbf{y}|) = |\mathbf{y}|_{\downarrow}$.

The proof is in Appendix B. We get from Proposition 2.3 that the difficulty in computing the prox of a sorted penalty at any point $\mathbf{y} \in \mathbb{R}^p$ is the same as computing this prox for sorted positive vectors $\mathbf{y} \in \mathcal{K}_p^+$ only. Moreover, it turns out that the latter can be replaced by an equivalent non-negative isotonic[2] constrained problem, as stated by Proposition 2.4.

**Proposition 2.4** (Proximal operator of sorted positive vectors)**.** *Let* $\mathbf{y} \in \mathcal{K}_p^+$, *i.e.* $y_1 \geq \cdots \geq y_p \geq 0$. *Then:*

$$\mathrm{prox}_{\eta\Psi}(\mathbf{y}) = \underset{\mathbf{x} \in \mathcal{K}_p^+}{\arg\min} P_{\eta\psi}(\mathbf{x}; \mathbf{y}) , \tag{5}$$

*where*

$$P_{\eta\psi}(\mathbf{x}; \mathbf{y}) \triangleq \sum_{i=1}^{p} \psi(x_i, \lambda_i) + \frac{1}{2\eta} \|\mathbf{y} - \mathbf{x}\|^2 . \tag{6}$$

Note that this minimization problem depends solely on the *unsorted penalty* constrained to a *convex* set. To sum up, if one is able to solve the problem given in Equation (5), then from Proposition 2.3, one can recover the proximal operator of $\eta\Psi$ at any $\mathbf{y} \in \mathbb{R}^p$.

## 2.2 The PAV algorithm for isotonic convex problems

The PAV algorithm has been introduced by Best & Chakravarti (1990) to solve the isotonic regression problem, i.e. the orthogonal projection onto the isotonic cone $\mathcal{K}_p^+$. Best et al. (2000) have extended it to the minimization of separable convex functions over $\mathcal{K}_p^+$.

We give PAV pseudo code in Appendix C and briefly explain below how it operates. In simple terms, the PAV algorithm starts from the unconstrained optimal point and partitions the solutions into blocks (singletons at initialization). It then iterates through the blocks, merging them using *a pooling operation* when the monotonicity constraint is violated:

- *Forward pass:* PAV merges the current working block with its successor if monotonicity is violated.
- *Backward pass:* Once this forward pooling done, the current working block is merged with its predecessors as long as monotonicity is violated.

Finally, the additional non-negative condition is dealt with by taking the positive part of the solution as the end of the algorithm (i.e. projecting on the non-negative cone). PAV is of linear complexity when the pooling operation is done in $\mathcal{O}(1)$. The correctness proof of the PAV algorithm (Best et al., 2000, Theorem 2.5) consists in showing that the KKT conditions are satisfied when the algorithm ends. The PAV algorithm works for *any separable convex functions* as long as one knows the pooling operation to be done when updating the values.

---

[2]as in isotonic regression, to be understood as monotonic

# 3 Generalization to sorted weakly-convex penalties

In this section, we describe how the PAV algorithm can be used to efficiently compute the prox operator of many Sorted nonconvex sparse penalties, such as sorted SCAD, sorted MCP or sorted log-sum, leveraging their weak convexity.

**Assumption 1** (Weak convexity)**.** *A function $\psi$ is said to be $\mu$-weakly convex if $\psi + \frac{\mu}{2}\|\cdot\|^2$ is convex for some $\mu > 0$. We focus in this section on sorted penalties built from scalar weakly convex penalties: there exists some $\mu > 0$ such that for any $i \in [\![1, p]\!]$, the scalar penalty $\psi(\cdot; \lambda_i)$ is $\mu$-weakly convex.*

The next proposition shows that computing the proximal operator under Assumption 1 comes down to solving a convex problem.

**Proposition 3.1.** *If the sorted penalty $\Psi$ is built from a positive scalar penalty $\psi$ satisfying Assumption 1 for some $\mu > 0$, then, for $0 < \eta < \frac{1}{\mu}$, the unsorted objective $P_{\eta\psi}$ in Equation* (6) *is convex and the proximal mapping $\mathrm{prox}_{\eta\Psi}$ is a singleton on $\mathbb{R}^p$.*

*Proof.* The proximal mapping is nonempty for any $\mathbf{x} \in \mathbb{R}^p, \eta > 0$. Then $P_{\eta\psi}$ is clearly convex if $\eta < \frac{1}{\mu}$ and Assumption 1 holds. From Proposition 2.3 and Proposition 2.4, as the set of minimizers of a strongly convex function, $\mathrm{prox}_{\eta\Psi}(\mathbf{y})$ is a singleton for all $\mathbf{y} \in \mathbb{R}^p$. □

**Remark 3.2.** *Proposition 3.1 requires that the stepsize $\eta$ is smaller than a constant depending on the weak-convexity parameter of the penalty. Although it may seem restrictive, note that it is consistent with the condition for convergence of the PGD algorithm (Equation* (4)*) that imposes $\eta < \frac{1}{L}$ with a L-smooth datafit $f$.*

## 3.1 Computation of the proximal operator with a PAV algorithm

With the stepsize $\eta$ taken as $\eta < 1/\mu$, then the prox problem becomes a convex isotonic problem so the PAV algorithm can be used.

The pooling operation, which updates value on a block $B$, $\chi(B)$, is defined as:

$$\chi(B) \triangleq \underset{z \in \mathbb{R}_+}{\arg\min} \sum_{i \in B} \frac{1}{2\eta}(z - y_i)^2 + \psi(z; \lambda_i) \ . \tag{7}$$

Proposition 3.3 shows that, for sorted weakly convex penalties, the pooling operation given by Equation (7) can be written as a *scalar proximal operation*.

**Proposition 3.3** (Writing $\chi$ as a proximal operation)**.** *Under Assumption 1, for a block of successive indices $B$, the minimizer $\chi(B)$ defined by Equation* (7) *is the proximal point of an averaged scalar penalty evaluated at an averaged data point.*

$$\chi(B) = \mathrm{prox}_{\frac{\eta}{|B|} \sum_{i \in B} \psi(\cdot; \lambda_i)}(\bar{y}_B) \ , \tag{8}$$

*where we use the notation $\bar{x}_B$ as the average value of the vector $\mathbf{x}$ on the block $B$: $\bar{x}_B = \sum_{i \in B} x_i$. Besides, if $\psi(\cdot; \lambda) = \lambda_i \psi_0(\cdot)$, then we have a direct formulation of $\chi(B)$ as:*

$$\chi(B) = \mathrm{prox}_{\bar{\lambda}_B \eta \psi_0}(\bar{y}_B) \ . \tag{9}$$

*Proof.* Let $\mathbf{y} \in \mathbb{R}^p$ such that $\mathbf{y} = |\mathbf{y}|_\downarrow$ the point for which we want to compute $\mathrm{prox}_{\eta\Psi}(\mathbf{y})$. Let $B = [\![q, r]\!]$ be a block of consecutive indices. Then, $\chi(B)$ is the unique zero of the function $f_B$ defined as

$$f_B(z) = |B| \left[ \frac{1}{\eta}(z - \bar{y}_B) + \frac{\sum_{i \in B} \psi'(z; \lambda_i)}{|B|} \right]$$

$$= |B| \frac{\partial}{\partial z} \left[ \frac{1}{2\eta}(z - \bar{y}_B)^2 + \frac{\sum_{i \in B} \psi(z; \lambda_i)}{|B|} \right] \ .$$

So, $\chi(B)$ is the unique minimizer of the strongly convex function $z \mapsto \frac{1}{2\eta}(z - \bar{y}_B)^2 + \frac{\sum_{i \in B} \psi(z; \lambda_i)}{|B|}$, which gives the expected result. □

**Remark 3.4.** *For a standard non-sorted penalty, one would apply the proximal operator component-wise. Proposition 3.3 points out that the use of sorted penalties enforces the proximal operation to be done block-wise, which enforces coefficient grouping, as intended.*

**Example 3.5** (Sorted MCP). *For any $\gamma > 0$, $\lambda > 0$, the MCP penalty is defined on $\mathbb{R}_+ \times \mathbb{R}_+$ as:*

$$\psi : (z; \lambda) \mapsto \begin{cases} \lambda z - \frac{z^2}{2\gamma} & \text{if } z \leq \gamma\lambda \ , \\ \frac{\lambda^2 \gamma}{2} & \text{otherwise} \ . \end{cases}$$

*It is $\frac{1}{\gamma}$-weakly convex, so is $\Psi$. From Proposition 3.3, the solution $\chi(B)$ of Equation (7) is the zero of the continuous, strictly increasing affine function $f_B$:*

$$f_B : z \mapsto \frac{1}{\eta}|B|(z - \bar{y}_B) + \sum_{i \in B} \left( \lambda_i - \frac{z}{\gamma} \right)_+ \ .$$

*By denoting for every $i \in B = [\![q, r]\!]$, $v_i \triangleq [|B|\gamma - \eta(i - q + 1)]\lambda_i + \eta\sum_{j=q}^i \lambda_j$, we have an explicit expression for $\chi(B)$ that only depends on the position of $\bar{y}_B$ in the sequence $(v_i)_{i \in B}$:*

$$\chi(B) = \frac{\bar{y}_B - \frac{\eta}{|B|}\sum_{j=q}^i \lambda_i}{1 - \frac{i-q+1}{|B|}\frac{\eta}{\gamma}} \ ,$$

*with $i$ such that $|B|v_{i+1} \leq \bar{y}_B < |B|v_i$ (we denote $v_{q-1} = +\infty$, $v_{r+1} = 0$ and $\sum_{j=q}^{q-1} = 0$).*

**Example 3.6** (Sorted Log-Sum). *For any $\epsilon > 0$, $\lambda > 0$, the log-sum penalty is defined on $\mathbb{R}_+ \times \mathbb{R}_+$ as $\psi : (z; \lambda) \mapsto \lambda \log\left(1 + \frac{z}{\epsilon}\right)$. It is $\frac{\lambda}{\epsilon^2}$-weakly convex. So, $\Psi$ is $\frac{\lambda_1}{\epsilon^2}$- weakly convex. Then, for every $\eta < \frac{\epsilon^2}{\lambda_1}$, $\mathrm{prox}_{\eta\Psi}$ can be computed in $\mathcal{O}(n)$ using the PAV algorithm and updating each block using the known formula of the proximal operator of log-sum (Prater-Bennette et al., 2022).*

$$\chi(B) = \begin{cases} 0 & \text{if } \bar{y}_B \leq \frac{\bar{\lambda}_B}{\epsilon} \ , \\ \frac{1}{2}(\bar{y}_B - \epsilon) + \sqrt{\frac{1}{4}(\bar{y}_B + \epsilon)^2 - \bar{\lambda}_B} & \text{otherwise} \ . \end{cases}$$

**Links with the MM approach** Feng & Zhang (2019) proposed a majorization-minimization algorithm in which they first upper bound $\Psi$ by $\Psi_+ = \Psi + \Psi_-$, and then minimize the majorizing function with one step of proximal gradient descent, using PAVA to compute $\mathrm{prox}_{\eta\Psi_+}$. For weakly convex penalties, one can take $\Psi_-$ equal to a quadratic. The main difference with our work is that we show that prox can be *directly computed*, without relying on a majorization. We provide in Appendix I more details on Feng & Zhang (2019) approach as well as experimental comparison.

## 4 Extension to nonconvex Sorted $\ell_q$

This section addresses the more complex case of the sorted $\ell_q$ penalty with $q \in ]0, 1[$: it provides characterizations for local and global minimizers, and explains how PAV can be adapted to compute them.

The $\ell_q$ penalty is not weakly convex, so cannot be treated as in the previous section. While being more difficult to handle, several experimental studies (Li et al., 2018; Yuan et al., 2019) show that $\ell_q$ outperforms $\ell_1$ or SCAD penalties for specific inverse problems: its recurrent appearance in the scientific literature makes it worthy of interest. Among the $\ell_q$ penalties, $\ell_{1/2}$ and $\ell_{2/3}$ stand out as the two penalties whose proximal operator can be computed in closed form (Xu et al., 2012; Chen et al., 2016). Nevertheless, we treat a generic $q$.

For the $\ell_q$ penalty, Proposition 2.4 still holds and so our prox minimization problem is an isotonic regression problem:

$$\min_{\mathbf{x} \in \mathcal{K}_p^+} \frac{1}{2\eta}\|\mathbf{x} - \mathbf{y}\|^2 + \sum_{i=1}^p \lambda_i x_i^q \ .$$

Up to a scaling of the sequence $(\lambda_i)_{i \in [\![1,p]\!]}$, it rewrites:

$$(P_p): \min_{\mathbf{x} \in \mathcal{K}_p^+} \|\mathbf{x} - \mathbf{y}\|^2 + \sum_{i=1}^{p} \lambda_i x_i^q . \tag{10}$$

Before delving into Equation (10), we study the scalar case.

### 4.1   Background on the prox of scalar $\ell_q$

For $y \geq 0, \lambda > 0$, we consider the scalar proximal objective function of $\ell_q$,

$$F : z \mapsto (y - z)^2 + \lambda z^q. \tag{11}$$

**Proposition 4.1** (Local minimizers of $F$). *We denote $\tau$ the threshold defined as $\tau \triangleq \left[\frac{2-q}{1-q}\right] \left(\frac{\lambda q (1-q)}{2}\right)^{\frac{1}{2-q}}$.*

- *If $y < \tau$, then $F$ has only one minimizer which is $0$.*
- *If $y \geq \tau$, then $F$ has two local minimizers on $\mathbb{R}_+$ which are $0$ and another one that we denote $\rho^+(y, \lambda)$.*

**Proposition 4.2** (Minimum of $F$). *We denote $T$ the threshold defined as $T \triangleq \left[\frac{1}{2}\frac{2-q}{1-q}\right](\lambda(1-q))^{\frac{1}{2-q}}$. Then,*

$$\arg\min_{z \in \mathbb{R}_+} F(z) = \begin{cases} \{0\} & \text{if } y < T, \\ \{0, \rho^+(y, \lambda)\} & \text{if } y = T, \\ \{\rho^+(y, \lambda)\} & \text{if } y > T, \end{cases}$$

*where $\rho^+$ is the minimizer introduced in Proposition 4.1.*

The proofs of Propositions 4.1 and 4.2 can be found in Xu et al. (2012); Chen et al. (2016).

Since the thresholds defined in Propositions 4.1 and 4.2 are such that $T \geq \tau$, we can distinguish the following three cases (also described in Figure 6, Appendix E):

1. $y \leq \tau$: the unique global minimizer of $F$ on $\mathbb{R}_+$ is $0$.
2. $\tau \leq y \leq T$: there are two local minimizers, $0$ and $\rho^+$, $0$ being the global minimizer of $F$ on $\mathbb{R}_+$.
3. $T < y$ : there are two local minimizers, $0$ and $\rho^+$, $\rho^+$ being the global minimizer of $F$ on $\mathbb{R}_+$.

**Lemma 4.3.** *For $y \geq \tau$, then $\rho^+$ the* local *minimizer of $F$ on $\mathbb{R}_{+*}$ satisfies the following inequality.*

$$\rho^+(y, \lambda) \geq \left(\frac{\lambda q (1-q)}{2}\right)^{\frac{1}{2-q}} .$$

**Lemma 4.4.** *For $y \geq T$, then $\rho^+$ the* global *minimizer of $F$ on $\mathbb{R}_+$ satisfies the following inequality.*

$$\rho^+(y, \lambda) \geq (\lambda(1-q))^{\frac{1}{2-q}} .$$

An extensive study of the proximal operator of $\ell_q$ is provided in Appendix D, as it sheds light on the nature of the functions to be minimized in the sorted case.

### 4.2   Local minimizers of the sorted $\ell_q$ prox problem

Our first goal is to describe the set of *local minimizers* of the problem $(P_p)$. We have the following characterization of these elements.

For a block $B = [\![q, r]\!]$, we define as $F_B$, or alternatively $F_{q:r}$ the *scalar* prox objective function on $\mathbb{R}_+$:

$$F_B = F_{q:r} : z \mapsto \sum_{i \in B} (y_i - z)^2 + \lambda_i z^q . \tag{12}$$

**Theorem 4.5** (Characterization of local minimizers). *We denote $\mathcal{L}_p$ the set of local minimizers of the problem $(P_p)$. Then, $\mathbf{u} \in \mathcal{L}_p$ if and only if, on each maximal[3] block $B = [\![q, r]\!]$ where $\mathbf{u}$ is constant equal to $\tilde{u}$, the two following results hold:*

*(i) $\tilde{u} \in \{\chi(B), 0\}$, where*

$$\chi(B) \triangleq \begin{cases} \rho^+(\bar{y}_B, \bar{\lambda}_B) & \text{if } \bar{y}_B \geq \tau_B \ , \\ 0 & \text{otherwise} \ . \end{cases} \tag{13}$$

*(ii) when $\tilde{u} = \chi(B)$, then for all $j \in [\![q, r-1]\!]$ we have,*

$$\text{either } \chi([\![q,j]\!]) \leq \chi(B) \leq \chi([\![j+1,r]\!]), \tag{14}$$
$$\text{either } \chi([\![q,j]\!]) \geq \chi([\![j+1,r]\!]) \geq \chi(B). \tag{15}$$

*Moreover if $\chi(B) > 0$, $F_{q:j}$ is increasing at $\chi(B)$ and $F_{j+1:r}$ is decreasing at $\chi(B)$ while if $\chi(B) = 0$ both $F_{q:j}$ and $F_{j+1:r}$ are increasing at $\chi(B)$.*

*Proof.* The detailed proof is in Appendix E; it relies on a careful study of the different possible variation profiles for $F_B$. Informally, the main ideas behind of the proof are as follows.

(i) Theorem 4.5 (i) is obtained from the fact that the value of each block of a local minimizer has to be itself a local minimizer of the corresponding subproblem.

(ii) Theorem 4.5 (ii) is obtained from the fact that any feasible infinitesimal move around a local minimizer should not decrease the objective value.

The second point of Theorem 4.5 highlights a key difference with the weakly convex case and the use of the PAV algorithm. In the weakly convex setting, PAV algorithm merges blocks that violate the isotonic constraint. Yet, for non weakly convex functions, Equation (15) points out a more pathological behavior: merging correctly ordered blocks, which PAV does not do, might actually lead to better solutions. Nevertheless, making use of the sufficient conditions of Theorem 4.5, we are able to show that, while being non exhaustive, the PAV algorithm still reaches a local minimizer of the proximal sorted $\ell_q$ problem.

**Theorem 4.6** (Convergence of PAV algorithm). *The PAV Algorithm 2 based on the update rule $B \mapsto \chi(B)$ with $\chi$ defined in Equation (13) finds a* local *minimizer of the problem $(P_p)$.*

*Proof.* Proof is given in Appendix F; it comes down to showing each pooling operation in the PAV Algorithm 2 preserves Equation (14) which is a sufficient condition to reach local optimality.

## 4.3 On global minimizers of the sorted $\ell_q$ prox problem

The goal of this section is to better describe *global* minimizers of the $\ell_q$ prox problem. To this end, we restrict ourselves to sequences of regularization parameters which write as $\lambda_i = \Lambda(i)$ with $\Lambda$ concave, non-decreasing and non-negative. In particular, this setting encompasses the famous OSCAR model (Bondell & Reich, 2008) where the regularization parameters $\lambda_i$ decrease linearly:

$$\lambda_i = \lambda_0 + \lambda(p - i) \quad \forall i \in [\![1, p]\!] \ . \tag{16}$$

**Theorem 4.7** (Necessary conditions on global minimizers). *Let the regularization parameters $(\lambda_i)_i$ be described by a concave, non-decreasing and non-negative function of $i \in \mathbb{N}$. Let $\mathbf{x}^p$ be a global minimizer of $(P_p)$. Then, $\mathbf{x}^p \in \mathcal{S}_p$, where $\mathcal{S}_p$ is given by*

$$\mathcal{S}_p \triangleq \left\{ (\mathbf{u}^{i^*-1}, \chi([\![i^*, p]\!]), \ldots, \chi([\![i^*, p]\!])), \mathbf{z}^1, \ldots, \mathbf{z}^p \right\} \ ,$$
$$\mathbf{z}^i \triangleq (\mathbf{u}^{i-1}, 0, \ldots, 0) \quad \forall i \in [\![1, p]\!] \ ,$$
$$\mathbf{u}^i \in \mathcal{L}_i \quad \forall i \in [\![1, p]\!] \ .$$

*where $i^*$ denotes the largest index such that $(\mathbf{u}^{i^*-1}, \chi([\![i^*, k]\!]), \ldots, \chi([\![i^*, k]\!]))$ is feasible.*

---

[3]meaning that if $B$ is extended, $\mathbf{x}$ is no longer constant on it

*Proof.* Proof of Theorem 4.7 is given in Appendix G. Restricting the sequence of regularization parameters restricts the set of points where Equation (15) occurs. This simplifies the setting, as the last block of the solution, which is always a *global* minimizer for its related *scalar* block function, can only be obtained by merging unsorted blocks, which is more intuitive. Then, the proof establishes these necessary conditions on *global* minimizers through a geometrical approach. It constructs other local minimizers using sufficient conditions outlined in Theorem 4.5, and show they can not achieve global optimality.

The importance of the necessary conditions stated in Theorem 4.7 is twofold. First, it gives more information on the structure of the last block. Then, it also highlights the need to consider separately the solutions built with a last block of zeros. It motivates the use of a modified version of the PAV algorithm, described below, which explicitly explores these cases.

**The D-PAV algorithm** The standard PAV algorithm iterates over the indices $k \in [\![1, p]\!]$. At each step $k$, it takes the result of size $k-1$, i.e. $\mathbf{u}^{k-1}$ and test if $(\mathbf{u}^{k-1}, \chi(\{k\}))$ is feasible. If not, it backtracks by merging the last component with the previous blocks until it gets $(\mathbf{u}^{k-1}_{[i-1]}, \chi([\![i, k]\!]), \dots, \chi([\![i, k]\!]))$ feasible. Doing so, it exactly computes $i^*$ the largest index such that the vector $(\mathbf{u}^{i^*-1}, \chi([\![i^*, k]\!]), \dots, \chi([\![i^*, k]\!])) \in \mathcal{K}_k$ and $\mathbf{u}^{i^*-1} \in \mathcal{L}_{i^*-1}$ (Theorem 4.6). Yet, it does not explore local minimizers with a last block of 0 (denoted as $\mathbf{z}$ in Theorem 4.7). This suggests introducing a decomposed PAV algorithm (D-PAV, Algorithm 1) to explicitly consider these local minimizers: as we compute the PAV solution, we store the intermediate solutions of size $k \le p$, complete them with zeros at the end of the vector, and finally keep the one leading to the smallest objective value.

---

**Algorithm 1:** D-PAV for sorted $\ell_q$

**input :** data $\mathbf{y} \in \mathbb{R}^p$, regularization sequence $(\lambda_i)_{i \in [\![1, k]\!]}$
$\mathbf{y} \leftarrow |\mathbf{y}|_\downarrow$
**for** $k \in [\![1, p]\!]$ **do**
  $\mathbf{x}^k_{1:k} \leftarrow \text{PAV}(\mathbf{y}_{[k]}, \boldsymbol{\lambda}_{[k]})$;                               /* Algorithm 2 */
  $\mathbf{x}^k_{i+1:p} \leftarrow 0$
  **if** $F(\mathbf{x}^k) < F(\mathbf{x})$ **then**
    | $\mathbf{x} \leftarrow \mathbf{x}^k$
  **end**
**end**
**return x**

---

### 4.4 Toward a generic algorithm ?

Our investigation of Sorted $\ell_q$ provides computational means along with theoretical guarantees for this specific penalty. Yet, our approach may be generalized: it also adapts to $\mu$ weakly-convex penalties in the *non weakly-convex regime* when assuming larger step sizes $\eta$ in the prox computation, i.e. $\eta \ge \frac{1}{\mu}$. In this setting, the prox objective shows similar variations to the case of the $\ell_q$ penalty (see Figure 6 for a reminder):

1. the existence of two thresholds, $\tau$ and $T$ such that
   (a) $\tau$ decides whether there is one minimizer which is 0 or two local minimizers, 0 and $\rho^+$ on $\mathbb{R}_+$.
   (b) $T \ge \tau$ decides whether 0 or $\rho^+$ is the global minimizer.
2. some lower bounds on the second minimizer $\rho_+$ which is non-decreasing with $\lambda$ such as in Lemma 4.3.

Given the knowledge of these two elements, we directly recover the necessary and sufficient conditions on local minimizers (Theorem 4.5).

As an illustration, we lay down the directions to adapt our results on $\ell_q$ to the log-sum penalty

$$\psi_\epsilon(x; \lambda) = \lambda \log\left(1 + \frac{|x|}{\epsilon}\right) , \tag{17}$$

where $\epsilon > 0$.

Prater-Bennette et al. (2022) derive explicit expressions for its proximal operator. In particular, they show the existence of two regimes depending on $\lambda, \epsilon$:

1. *Weakly-convex regime:* As seen in Section 3, the log-sum penalty is $\frac{\lambda}{\epsilon^2}$-weakly convex. So, for $\eta < \frac{\epsilon^2}{\lambda_1}$, the proximal operator $\text{prox}_{\eta \Psi_\epsilon}$ can be computed using standard PAV algorithm.

2. *Not weakly-convex regime:* Prater-Bennette et al. (2022, Lemma 2, Proposition 2) give the existence of the two thresholds $\tau$ and $T$:

   (a) $\tau = 2\sqrt{\eta\lambda} - \epsilon$ decides whether there is one minimizer which is 0 or two local minimizers, 0 and $\rho^+$ on $\mathbb{R}_+$.
   (b) $T \geq \tau$, defined implicitly, decides whether 0 or $\rho^+$ is the global minimizer.

   Prater-Bennette et al. (2022, Lemma 3) also explicit a lower bound on $\rho^+$, which is non-decreasing with $\lambda,$ : for $y \geq \tau$, then $\rho^+ \geq \sqrt{\eta\lambda} - \epsilon$.

The D-PAV algorithm can be used in this setting, leveraging on the explicit form of the proximal operator given by Prater-Bennette et al. (2022).

# 5 Experiments

The code to reproduce all the experiments is included in the supplementary material for frictionless reproducibility. The attached code package relies on `sklearn` and `skglm` (Pedregosa et al., 2011; Bertrand et al., 2022); it will be released on GitHub upon acceptance.

## 5.1 Denoising

In this first experiment, we demonstrate the property that, generalizing from classical $\ell_1$ versus nonconvex penalties, one can expect from *sorted* nonconvex penalties: they lead to estimators with less bias.

The setup is the following: we generate a true vector $\mathbf{x}^* \in \mathbb{R}^{28}$ with 4 equal-sized clusters of coefficients, on which it takes the values $7, -5, 3$ and -1. The clusters are taken contiguous simply to visualize them better in Figure 1, but we stress that this does not help the algorithms: their performance would be the same if we shuffled the vector to be recovered. We add i.i.d Gaussian noise of standard deviation 0.3 to $\mathbf{x}$ to create 1000 samples of noisy vector $\mathbf{x}$.

To denoise $\mathbf{x}$, we apply the proximal operators of SLOPE, sorted MCP and sorted $\ell_{0.5}$ respectively. The sequence $(\lambda_i)$ is selected[4] as $\lambda_i = r(28 - i)$, with $r > 0$ controlling the strength. For each of the 3 penalties, we compute denoised versions of $\mathbf{x}$ by applying to it their proximal operator, for 100 geometrically-spaced values of $r$. For each resulting denoised vector $\hat{\mathbf{x}}$, we define the normalized error as $\|\hat{\mathbf{x}} - \mathbf{x}^*\|/\|\mathbf{x}^*\|$. To measure the cluster recovery performance, we define the binary matrix $M(\mathbf{x})$ as having entry $i, j$ equal to 1 if $|x_i| = |x_j|$ (i.e. $i$ and $j$ are in the same cluster in $\mathbf{x}$) and 0 otherwise. The reported F1 score is then the F1 score between $M(\hat{\mathbf{x}})$ and $M(\mathbf{x}^*)$, i.e. the F1 score of the classification task "predict if two entries of the vector belong to the same cluster". An F1 score of 1 means that $\hat{\mathbf{x}}$ perfectly recovers the clusters of $\mathbf{x}^*$.

Figure 2 displays the variation of F1 score and error. For each penalty, we define the optimal penalty strength $r$ as the lowest $r$ for which the F1 score is superior to 80% of the maximal F1 score reached for this penalty (vertical dashed lines). One can see that, for an equivalent F1 score, nonconvex sorted penalty are less biased and reach a lower error. Denoised vector for the optimal choice of $r$ are displayed in Figure 1.

## 5.2 Regression

We now turn to the setting of Equation (2), with a least squares datafit $f(\mathbf{x}) = \frac{1}{2}\|\mathbf{A}\mathbf{x} - \mathbf{b}\|^2$. The setup is the following. The matrix $\mathbf{A}$ is in $\mathbb{R}^{50 \times 100}$, it is random Gaussian with Toeplitz-structured covariance, equal to $(0.98^{|i-j|})_{i,j}$. The ground truth $\mathbf{x}^*$ has entries from 1 to 10 and 50 to 55 equal to 5, the rest being 0. The target $\mathbf{b}$ is $\mathbf{A}\mathbf{x}^* + \boldsymbol{\epsilon}$ where $\boldsymbol{\epsilon}$ has independent Gaussian entries, scaled so that the SNR $\|\mathbf{A}\mathbf{x}^*\|/\|\boldsymbol{\epsilon}\|$ is equal to

---

[4]for $\ell_{0.5}$ we use $r(28 - i)^{1.5}$ as concave sequences empirically turned out to yield better performance for $\ell_q$

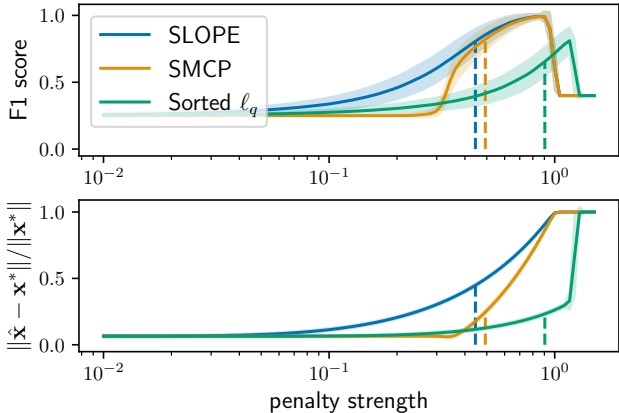

Figure 2: *Top*: F1 score (averaged on the 1000 samples, with standard deviation) for cluster recovery, as a function of the penalty strength. *Bottom*: Normalized estimation error (averaged on the 1000 samples, with standard deviation), as a function of the penalty strength. For each penalty, the dashed line is set at 80% of the maximum F1 score and displays a tradeoff between small estimation error and high F1 score.

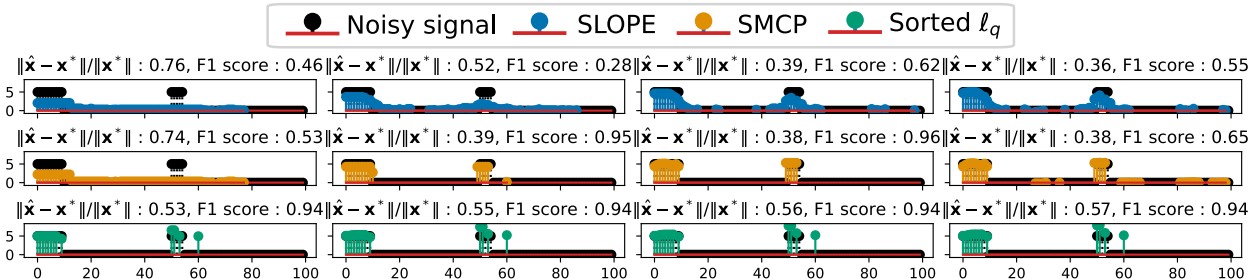

Figure 3: Solutions of $\Psi$-penalized regression for SLOPE (blue), sorted MCP (yellow) and sorted $\ell_{1/2}$ (green), for decreasing penalty strengths (from left column to right). The ground truth is in black; SLOPE has more false positive and bias.

5. All penalties use the Benjamini-Hochberg sequence $\lambda_j = r \operatorname{prob}(1 - 0.1j/(2p))$ where prob is the probit function, and $r > 0$ a scaling parameter to control penalty strength.

Figure 3 displays the result for 4 values of penalization strength. It highlights the bias of SLOPE, which creates many false positives and still underestimates the ground truth strength as the penalty strength diminishes. On the contrary, sorted $\ell_{1/2}$ and MCP have few false positives and better identify the magnitude of the true nonzero coefficient.

### 5.3 Discussion on the D-PAV algorithm for Sorted $\ell_q$

We introduced Section 4.3 the D-PAV algorithm, a variation of the PAV algorithm to handle Sorted $\ell_q$. This section highlights when and how D-PAV succeeds where standard PAV algorithm would fail. We recall that D-PAV specifically consider solutions that have a block of zeros at the end.

**D-PAV algorithm versus standard PAV algorithm.** We recall that the pooling operation of the PAV algorithm consists in solving the *scalar* objective function taken on each block $B$. In the case of $\ell_q$, we have seen that there exist two thresholds $\tau$ and $T$ which respectively state if one or two local minima exist / which one is the global minima. Hence, in order to use PAV in this non-convex setting (yet loosing convergence guarantees), there would be two ways to define the pooling operation:

- setting it to $\chi(B)$ defined in Equation (13), i.e, the largest local minimizer of $F_B$,
- setting it to $\chi^*(B) \in \{\chi(B), 0\}$, the global minimizer of $F_B$.

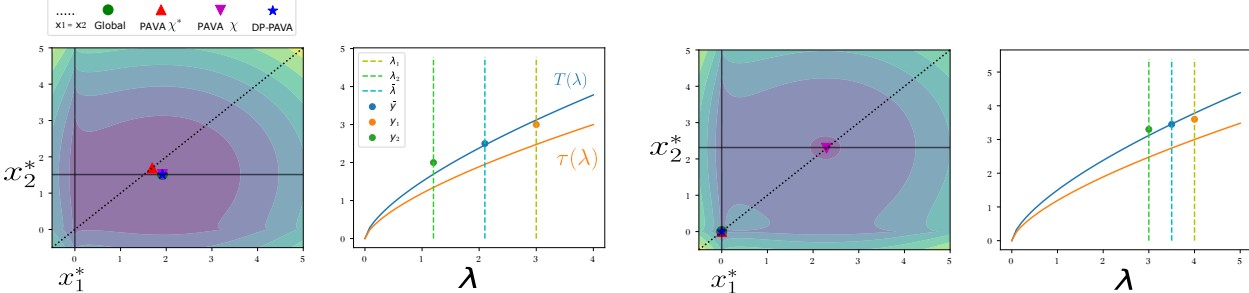

Figure 4: LEFT: Landscape of the proximal objective of $\ell_{1/2}$ and solutions for different algorithms. RIGHT: Positions of $(y, \lambda)$ w.r.t. thresholds $T$ and $\tau$.

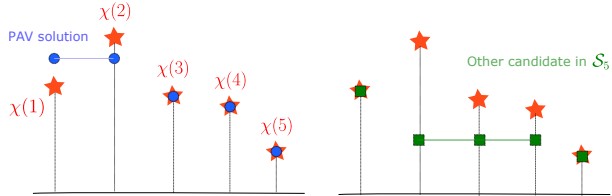

Figure 5: LEFT: The PAV algorithm merges unsorted blocks. RIGHT: The vector in green has been merged using Equation (15) and is in $\mathcal{S}_p$: hence, it may be a global minimizer.

In Figure 4, we show that these two ideas (referred respectively to PAV$_\chi$ and PAV$_{\chi^*}$) both fail in finding the global solution in simple 2D toy examples. In contrast, the proposed D-PAV algorithm (see Section 4.3) succeeds. The setting we considered here is the following. For both, $y_1 \geq \tau_1$ and $y_2 \geq \tau_2$ so $F_1$ and $F_2$ have both two local minimizers. Besides, we also have $y_1 < T_1$ and $y_2 > T_2$ meaning $F_1$ has 0 as global minimizer and $F_2$ has $\chi(2)$ as its global minimizer.

**Building counter-examples to D-PAV ?** We recall that Theorem 4.7 defines a subset of local minimizers which contain the global minimizer. While the D-PAV algorithm explores this subset, there may exist some setting where the algorithm misses some candidates and fail to recover the global minimizer. From the form of the solution, we know we at least recover the correct last block. Yet, on the first part of the solution, one may be able to find other blocks structure by merging correctly ordered blocks (Equation (15)). The following experiments show that the combination of $(\mathbf{y}, \boldsymbol{\lambda})$ such that this setting may occur are very rare. Even by purposely constructing such adversarial examples, we cannot find a setting where D-PAV fails.

*D-PAV versus exhaustive search.* To benchmark our D-PAV approach, we compare it with the bruteforce approach where we compute all the possible partitions of the vector into blocks. We take $\mathbf{y} \in \mathbb{R}^{10}$: each point $y_i$ is such that $y_i = T(\lambda_i) + \epsilon$ with $\epsilon \sim \mathcal{N}(-0.3, 1)$. We compute the value of the blocks using $\chi$. We test on 10 different seeds for which D-PAV always recovers the solution. We also compare ourselves for $\mathbf{y} \in \mathbb{R}^{100}$ with the Sequential Least Squares Programming implemented in the `scipy` library: we run on 100 initial points and keep the better solution. Once again, D-PAV systematically beats the solution from `scipy`.

*Adversarial settings.* The D-PAV algorithm could fail in a setup where it does not merge sorted blocks $\chi(B_1) \geq \chi(B_2)$ while $\chi(B_1 \cup B_2)$ could have achieved a better objective value. See Figure 5 for an illustration. To explore the rate of occurence of such a setup, we do the following experiment. We fix a value for $(y_{B_2}, \lambda_{B_2})$. We are looking for a set of points such that $y_{B_1} \geq y_{B_2}$, $\lambda_{B_1} \geq \lambda_{B_2}$ and $\chi(B_1) \geq \chi(B_2) \geq \chi(B_1 \cup B_2)$. As, $\chi(B_1 \cup B_2)$ is a function of the averaged $(y, \lambda)$, it rewrites as a function of $(y_{B_1 \cup B_2} = ty_{B_1} + (1 - t)y_{B_2}, \lambda_{B_1 \cup B_2} = t\lambda_{B_1} + (1 - t)\lambda_{B_2})$. Then, we look for $(y_{B_1}, \lambda_{B_1})$ and $t$ such that $\chi(B_1) \geq \chi(B_2) \geq \chi(B_1 \cup B_2)$. Results are plotted in Figure 9: we observe that the set of points where D-PAV could theorically fail (indeed, it nevers merges $B_1$ and $B_2$ if $\chi(B_1) \geq \chi(B_2)$) is scarce. Next, we pick such values of $(y, \lambda)$ and build a *candidate* counter-example in $\mathbb{R}^6$ as in Figure 5. Our D-PAV algorithm systematically yields better solutions than the ones obtained with the merging described above.

## 6 Conclusion

Sorted nonconvex penalties generalize the Sorted $\ell_1$ norm penalty: they benefit from both the sparsity-enhancing and clustering properties of SLOPE but promote solutions with less bias. We have proposed a unified approach to deal with regularized regression problems using such penalties by providing efficient algorithms to compute, exactly or approximately, their proximal operator, which paves the way for a more general use in practical cases.

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

## A  Non-convex penalties

We give here a few examples of non-convex penalties used as alternatives to the $\ell_1$ norm to mitigate amplitude bias of the solution.

**Definition A.1** (Minimax Concave Penalty (MCP), Zhang et al. (2013))**.** *For any $\gamma > 0$, $\lambda > 0$, the MCP is defined by $\psi(|\cdot|; \lambda)$ where, for all $t \geq 0$,*

$$\psi(t; \lambda) = \begin{cases} \lambda t - \frac{t^2}{2\gamma} & \text{if } t \leq \gamma\lambda \ , \\ \frac{\lambda^2 \gamma}{2} & \text{otherwise.} \end{cases} \tag{18}$$

**Definition A.2** (Smoothly Clipped Absolute Deviation (SCAD), Fan & Li (2001))**.** *For any $\gamma > 2$, $\lambda > 0$, the SCAD is defined by $\psi(|\cdot|; \lambda)$ where, for all $t \geq 0$,*

$$\psi(t; \lambda) = \begin{cases} \lambda t & \text{if } t \leq \lambda \ , \\ \frac{2\gamma\lambda t - t^2 - \lambda^2}{2(\gamma-1)} & \text{if } \lambda < t \leq \gamma\lambda \ , \\ \frac{\lambda^2(\gamma+1)}{2} & \text{otherwise.} \end{cases} \tag{19}$$

**Definition A.3** (Log-sum penalty, Candes et al. (2008))**.** *For any $\epsilon > 0$, $\lambda > 0$, the log-sum penalty is defined by $\psi(|\cdot|; \lambda)$ where, for all $t \geq 0$,*

$$\psi(t; \lambda) = \lambda \log\left(1 + \frac{t}{\epsilon}\right) \ . \tag{20}$$

**Definition A.4** ($\ell_q$ pseudo-norm)**.** *For $0 < q < 1$, $\lambda > 0$, the $\ell_q$ pseudo-norm regularizer is, for all $t \geq 0$:*

$$\psi(t; \lambda) = \lambda t^q \ . \tag{21}$$

# B    Properties of sorted penalties

We recall several properties of proximal operators of sorted penalties, that will ease their computation, and give the proof of Propositions 2.3 and 2.4. Note that these properties have already been shown in several previous works Zeng & Figueiredo (2014); Bogdan et al. (2015); Feng & Zhang (2019). First, to ease the presentation of the following proofs, let introduce the objective function of the sorted proximal operator

$$Q_{\eta\Psi}(\mathbf{x}; \mathbf{y}) \triangleq \frac{1}{2\eta}\|\mathbf{y} - \mathbf{x}\|_2^2 + \Psi(\mathbf{x}) = \frac{1}{2\eta}\|\mathbf{y} - \mathbf{x}\|_2^2 + \sum_{i=1}^p \psi(|x_{(i)}|, \lambda_i), \tag{22}$$

which must not be confused with the non-sorted objective $P_{\eta\psi}$ defined in Equation (6).

**Proposition B.1** (Sign of the proximal operator). *Let $\mathbf{x} \in \mathbb{R}^p$ and $\mathbf{p} \in \mathrm{prox}_{\eta\Psi}(\mathbf{x})$. Then, for every $i \in [\![1, p]\!]$:*

$$p_i x_i \geq 0 \ .$$

*Proof.* Suppose that there exists $i \in [\![1, p]\!]$ such that $x_i p_i < 0$. Since for every $\mathbf{x} \in \mathbb{R}^p$, $\Psi(\mathbf{x}) = \Psi(|\mathbf{x}|)$, we can assume $x_i > 0$ and $p_i < 0$ without loss of generality. It then follows that:

$$(x_i - p_i)^2 > (x_i - |p_i|)^2.$$

Let $\tilde{\mathbf{p}}$ be equal to $\mathbf{p}$ everywhere except at index $i$ where $\tilde{p}_i = |p_i|$. One has $\Psi(\tilde{\mathbf{p}}) = \Psi(\mathbf{p})$ and:

$$\begin{aligned}
Q_{\eta\Psi}(\tilde{\mathbf{p}}, \mathbf{x}) &= \frac{1}{2}\sum_{j \neq i}(x_j - p_j)^2 + \frac{1}{2}(x_i - |p_i|)^2 + \eta\Psi(\tilde{\mathbf{p}}) \\
&< \frac{1}{2}\sum_{j \neq i}(x_j - p_j)^2 + \frac{1}{2}(x_i - p_i)^2 + \eta\Psi(\tilde{\mathbf{p}}) \\
&= \frac{1}{2}\sum_{j \neq i}(x_j - p_j)^2 + \frac{1}{2}(x_i - p_i)^2 + \eta\Psi(\mathbf{p}) = Q_{\eta\Psi}(\mathbf{p}, \mathbf{x}),
\end{aligned}$$

which contradicts the optimality of $\mathbf{p}$.    □

**Proposition B.2** (Monotonicity of the proximal operator). *Let $\mathbf{x} \in \mathbb{R}^p$ and $\mathbf{p} \in \mathrm{prox}_{\eta\Psi}(\mathbf{x})$. Then, the components of $\mathbf{x}$ and those of $\mathbf{p}$ are ordered in the same way: $\forall i, j \in [\![1, p]\!], x_i < x_j \implies p_i \leq p_j$.*

*Proof.* Let $i, j \in [\![1, p]\!]$ such that $x_i < x_j$ and $p_i > p_j$. Then:

$$(x_i - p_i)^2 + (x_j - p_j)^2 = x_i^2 + x_j^2 + p_i^2 + p_j^2 - 2x_i p_i - 2x_j p_j.$$

Yet by assumption $(x_i - x_j)(p_j - p_i) > 0$, and so $-x_i p_i - x_j p_j > -x_j p_i - x_i p_j$. It follows:

$$(x_i - p_i)^2 + (x_j - p_j)^2 > x_i^2 + x_j^2 + p_i^2 + p_j^2 - 2x_j p_i - 2x_i p_j = (x_j - p_i)^2 + (x_i - p_j)^2.$$

Let $\tilde{\mathbf{p}}$ be equal to $\mathbf{p}$, except for $\tilde{p}_i = p_j$ and $\tilde{p}_j = p_i$. As $\Psi$ is a sorted penalty, $\Psi(\mathbf{p}) = \Psi(\tilde{\mathbf{p}})$. Therefore,

$$\begin{aligned}
Q_{\eta\Psi}(\tilde{\mathbf{p}}, \mathbf{x}) &= \frac{1}{2}\sum_{k \neq i, j}(x_k - p_k)^2 + \frac{1}{2}(x_j - p_i)^2 + \frac{1}{2}(x_i - p_j)^2 + \eta\Psi(\tilde{\mathbf{p}}) \\
&< \frac{1}{2}\sum_{k \neq i, j}(x_k - p_k)^2 + \frac{1}{2}(x_i - p_i)^2 + \frac{1}{2}(x_j - p_j)^2 + \eta\Psi(\mathbf{p}) = Q_{\eta\Psi}(\mathbf{p}, \mathbf{x}),
\end{aligned}$$

which contradicts the optimality of $\mathbf{p}$.    □

*Proof of Proposition 2.3.* For any $\mathbf{y} \in \mathbb{R}^p$, we know from Proposition B.1 that $\mathbf{y}$ and $\text{prox}_{\eta\Psi}(\mathbf{y})$ share the same sign element-wise. As flipping the sign of an element of $\mathbf{y}$ does not change the value of $\Psi(\mathbf{y})$, it follows easily that $\text{prox}_{\eta\Psi}(\mathbf{y}) = \text{sign}(\mathbf{y}) \odot \text{prox}_{\eta\Psi}(|\mathbf{y}|)$. It remains to show that for $\mathbf{y}$ non-negative:

$$\text{prox}_{\eta\Psi}(\mathbf{y}) = \mathbf{P}_{\mathbf{y}}^{\top} \, \text{prox}_{\eta\Psi}(\mathbf{P}_{|\mathbf{y}|}\mathbf{y}). \tag{23}$$

For any permutation matrix $\mathbf{Q}$ and any $\mathbf{x}, \mathbf{y} \in \mathbb{R}^p$, $\|\mathbf{Q}(\mathbf{y} - \mathbf{x})\|^2 = \|\mathbf{y} - \mathbf{x}\|^2$. Moreover, by definition of the sorted penalty $\Psi$, we have $\Psi(\mathbf{P}_{|\mathbf{y}|}\mathbf{y}) = \Psi(\mathbf{y})$. Thus for any $\mathbf{x}, \mathbf{y} \in \mathbb{R}^p$:

$$\Psi(\mathbf{P}_{|\mathbf{y}|}\mathbf{y}) + \frac{1}{2}\|\mathbf{P}_{|\mathbf{y}|}(\mathbf{y} - \mathbf{x})\|^2 = \Psi(\mathbf{y}) + \frac{1}{2}\|\mathbf{y} - \mathbf{x}\|^2 \ ,$$

from which we deduce Equation (23), by change of variable $\mathbf{y}' = \mathbf{P}_{|\mathbf{y}|}\mathbf{y}$. □

*Proof of Proposition 2.4.* First, let $\mathbf{x}^* \in \text{prox}_{\eta\Psi}(\mathbf{y})$. From Propositions B.1 and B.2, because $\mathbf{y}$ is sorted and has positive entries, so is $\mathbf{x}^*$: $\mathbf{x}^* = |\mathbf{x}^*|_{\downarrow}$. Thus,

$$\text{prox}_{\eta\Psi}(\mathbf{y}) = \underset{\mathbf{x} \in \mathcal{K}_p^+}{\arg\min} \, Q_{\eta\Psi}(\mathbf{x}, \mathbf{y}).$$

Then, given that for every $\mathbf{x} \in \mathcal{K}_p^+$, $Q_{\eta\Psi}(\mathbf{x}, \mathbf{y}) = P_{\eta\Psi}(\mathbf{x}, \mathbf{y})$ completes the proof. □

## C  Pool Adjacent Violators algorithm

In the following, a block is a set of consecutive indices of $[\![1, p]\!]$ on which the components of the iterate $\mathbf{x}$ are equal. At each iteration of the algorithm, a block $B_0 \triangleq [\![q, r]\!]$ is taken as the *working block*. Likewise, $B_- \triangleq [\![..., q-1]\!]$ and $B_+ \triangleq [\![r+1, ...]\!]$ respectively denote the blocks before and after $B_0$, potentially empty. We denote $x_B$ as the common value of the components of $\mathbf{x}$ on the block $B$. We introduce two functions that operate on blocks: $\mathbf{Pred}(B)$ (resp. $\mathbf{Succ}(B)$) returns the block just before (resp. after) the block $B$ if it exists, otherwise, it returns the empty set.

---

**Algorithm 2:** PAV algorithm

**input :** data $\mathbf{y} \in \mathbb{R}^p$
$\mathbf{x} \leftarrow (\chi(\{1\}), \ldots, \chi(\{p\}))$ ;                    /* Initialisation with unconstrained solution */
$B_- \leftarrow \emptyset$
$B_0$ is the block such that $1 \in B_0$ and $B_+ \leftarrow \mathbf{Succ}(B_0)$
**while** $B_+ \neq \emptyset$ **do**
    **if** $x_{B_0} \geq x_{B_+}$ **then**
       |  $B_0 \leftarrow B_+$, $B_- \leftarrow B_0$, $B_+ \leftarrow \mathbf{Succ}(B_+)$
    **end**
    ;                                      /* If blocks are ordered, move forward */
    **else**
        Update $\mathbf{x}$ on $B_0 \cup B_+$: $x_{B_0 \cup B_+} \leftarrow \chi(B_0 \cup B_+)$ ;           /* If not, pooling operation */
        $B_0 \leftarrow B_0 \cup B_+$, $B_+ \leftarrow \mathbf{Succ}(B_0)$
        **while** $(x_{B_-} \leq x_{B_0}) \wedge (B_- \neq \emptyset)$ **do**
            Update $\mathbf{x}$ on $B_- \cup B_0$: $x_{B_- \cup B_0} \leftarrow \chi(B_- \cup B_0)$ ;   /* Backward pass to ensure correct
            ordering after pooling */
            $B_0 \leftarrow B_- \cup B_0$, $B_- \leftarrow \mathbf{Pred}(B_-)$
        **end**
    **end**
**end**
**return** $(\mathbf{x})_+$

---

## D   Proximal operator of $\ell_q$

We aim to find the minimizers of the function $F$ defined as:

$$F : z \mapsto (y - z)^2 + \lambda z^q$$

The function $F$ is defined on $\mathbb{R}_+$ and differentiable on $\mathbb{R}_{+*}$, thus its minimum may be attained at $0$ or at a critical point of $F$ on $\mathbb{R}_{+*}$. This motivates the following study of the variations of $F$. We denote by $f = F'$, defined on $\mathbb{R}_{+*}$.

$$f : z \mapsto 2 \left( z - y + \frac{\lambda q}{2} z^{q-1} \right)$$

**Proposition D.1.** *If $y > \tau \triangleq \left[ \frac{2-q}{1-q} \right] \left( \frac{\lambda q (1-q)}{2} \right)^{\frac{1}{2-q}}$, then $f$ admits exactly two zeros on $\mathbb{R}_{+*}$. We denote $\rho^+$ the largest one which locally minimizes $F$.*

*Proof.* The function $f$ is differentiable on $\mathbb{R}_{+*}$:

$$f'(z) = 2 \left( 1 - \frac{\lambda q (1-q)}{2} z^{q-2} \right)$$

We study the variations of $f$. We denote its unique minimum $m$.

$$f'(z) = 0 \iff z = \left( \frac{\lambda q (1-q)}{2} \right)^{\frac{1}{2-q}} \triangleq m$$

$$f(m) = 2 \left( m \left[ 1 + \lambda \frac{q}{2} m^{q-2} \right] - y \right) = 2 \left( m \left[ 1 + \lambda \frac{q}{2} \frac{2}{\lambda q (1-q)} \right] - y \right)$$
$$= 2 \left( m \frac{2-q}{1-q} - y \right) = 2 \left[ \left( \frac{2-q}{1-q} \right) \left( \frac{\lambda q (1-q)}{2} \right)^{\frac{1}{2-q}} - y \right]$$

| $x$ | $0$ | $m$ | $+\infty$ |
|-----|-----|-----|-----------|
| $f$ | $+\infty$ $\searrow$ | $f(m)$ | $\nearrow$ $+\infty$ |

- Either $f(m) > 0$ and $f$ is positive on $\mathbb{R}_{+*}$
- Either $f(m) = 0$ and $m$ is the only root of $f$.
- Either $f(m) < 0$ and $f$ has 2 roots: we denote $\rho^+$ the largest one.

The condition $f(m_B) < 0$ rewrites as follows.

$$y > \left( \frac{2-q}{1-q} \right) \left( \frac{\lambda q (1-q)}{2} \right)^{\frac{1}{2-q}} = \tau \tag{24}$$

As $f = F'$, it follows:

- Either $y \leq \tau$, and $F$ has no local minima on $\mathbb{R}_{+*}$,
- Either $y > \tau$, and $F$ has one local minima on $\mathbb{R}_{+*}$ which is $\rho^+$.

$\square$

*Proof of Proposition 4.2.* Assume 0 is the global minimizer of $F$ over $\mathbb{R}_+$, meaning $F(0) < F(z)$ for all $z > 0$. It translates as:

$$z^2 - 2yz + \lambda z^q > 0 \quad \forall z > 0$$

i.e. $z - 2y + \lambda z^{q-1} > 0$ for all $z > 0$. Then, we define $g : z \mapsto z + \lambda z^{q-1}$.

$$g'(z) = 1 + (q-1)\lambda z^{q-2}$$

$$g''(z) = (q-1)(q-2)\lambda z^{q-3} > 0 \; \forall z > 0$$

From the sign of $g''$, we have $g$ strictly convex on $\mathbb{R}_{+*}$. It achieves its minimum value at $M \triangleq (\lambda(1-q))^{\frac{1}{2-q}}$. Then, $g(z) > 2y$ for every $z > 0$ if and only if $g(M) > 2y$, which translates as $y < T$. $\qquad \square$

*Proof of Lemma 4.3.* Because $\rho^+$ is defined as the largest root of $f$ (the derivative of the prox objective function $F$), $\rho^+$ is larger than the minimum of $f$ which gives the inequality. $\qquad \square$

*Proof of Lemma 4.4.* We assume $y > T$ so $\rho^+$ is the global minimizer of $F$. Using the previous notations, we have $M \triangleq (\lambda(1-q))^{\frac{1}{2-q}}$. We compute $f(M)$:

$$
\begin{aligned}
f(M) &= 2\left(M - y + \frac{\lambda q}{2}M^{q-1}\right) \\
&= 2\left(M\left(1 + \frac{\lambda q}{2}M^{q-2}\right) - y\right) \\
&= 2\left(M\left(1 + \frac{\lambda q}{2\lambda(1-q)}\right) - y\right) \\
&= 2\left(M\left(\frac{2-q}{2(1-q)}\right) - y\right) \\
&= 2(T - y) < 0
\end{aligned}
$$

So, we get that $M < \rho^+$ as $f > 0$ on $(\rho^+, +\infty)$. $\qquad \square$

## E   Proof of Theorem 4.5

We first recall some important notations. If we consider a block of successive indices $B = [\![q, r]\!]$, We define as $F_B$ or $F_{q:r}$ the prox objective function on a block of successive indices $B$. The prox objective function is defined on $\mathbb{R}_+^p$ or on $\mathbb{R}_+$, depending if it is evaluated at a vector or scalar point:

$$F_B(\mathbf{z}) = \sum_{i \in B}(y_i - z_i)^2 + \lambda_i z_i^q \;,$$

$$F_B(z) = \sum_{i \in B}(y_i - z)^2 + \lambda_i z^q \;.$$

In similar fashion, we denote $f_B$ or $f_{q:r}$ the derivative of $F_B$ on $\mathbb{R}_+^p$ or on $\mathbb{R}_+$. We define, on a block $B$, $\chi(B)$ which is the largest largest local minimizer of $F$.

$$\chi(B) \triangleq \begin{cases} \rho^+(\bar{y}_B, \bar{\lambda}_B) & \text{if } \bar{y}_B \geq \tau_B \\ 0 & \text{otherwise} \;, \end{cases}$$

where $\rho^+$ and $\tau_B$ are defined in Proposition 4.1. We recall the three different shapes that (scalar) $F_B$ may have :

1. $\bar{y}_B \leq \tau_B$ : $\chi(B) = 0$ and is the unique global minimizer of $F_B$ on $\mathbb{R}_+$ (Figure 6a).

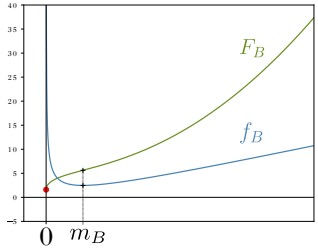
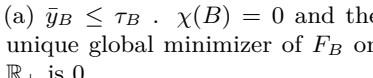
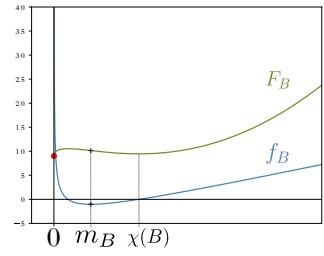
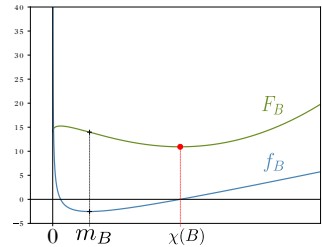

(a) $\bar{y}_B \leq \tau_B$ . $\chi(B) = 0$ and the unique global minimizer of $F_B$ on $\mathbb{R}_+$ is 0.

(b) $\tau_B \leq \bar{y}_B \leq T_b$. $\chi(B) = \rho^+(\bar{y}_B, \bar{\lambda}_B)$ and 0 is the global minimizer of $F_B$ on $\mathbb{R}_+$.

(c) $T_B < \bar{y}_B$: $\chi(B) = \rho^+(\bar{y}_B, \bar{\lambda}_B)$ and $\chi(B)$ is the global minimizer of $F_B$ on $\mathbb{R}_+$.

Figure 6: The 3 different possible shapes of $F_B$. The red dot represents the global minimizer of $F_B$.

2. $\tau_B \leq \bar{y}_B \leq T_b$ : $\chi(B) = \rho^+(\bar{y}_B, \bar{\lambda}_B)$ but $\chi(B)$ is not the global minimizer of $F_B$ on $\mathbb{R}_+$ (0 is) (Figure 6b).

3. $T_B < \bar{y}_B$ : $\chi(B) = \rho^+(\bar{y}_B, \bar{\lambda}_B)$ and $\chi(B)$ is the global minimizer of $F_B$ on $\mathbb{R}_+$ (Figure 6c).

In order to prove Theorem 4.5, we need the following lemma which shows that by merging two blocks that are not sorted in the correct order, the new value on the merged block is in between the two previous ones.

**Lemma E.1.** *Let $B_1$ and $B_2$ be two successive blocks of indices. Then, if $\chi(B_1) < \chi(B_2)$, the merged block satisfies the following inequality.*

$$\chi(B_1) \leq \chi(B_1 \cup B_2) \leq \chi(B_2) \ .$$

*Proof.* We denote $B \triangleq B_1 \cup B_2$ the merged block. We first study the case where $\chi(B_1) = 0$, then if $\chi(B) = 0$, the inequality trivially holds. Otherwise $\chi(B) = \rho^+(\bar{y}_B, \bar{\lambda}_B)$. Then, the two following equality holds:

$$f_B\left[\chi(B)\right] = f_{B_1}\left[\chi(B)\right] + f_{B_2}\left[\chi(B)\right] \quad \textit{(by separability)}$$
$$f_B\left[\chi(B)\right] = 0 \quad \textit{(by definition)} \ .$$

As $F_{B_1}$ is increasing on $\mathbb{R}_+$(Figure 6a), we have $f_{B_1}\left[\chi(B)\right] > 0$ so $f_{B_2}\left[\chi(B)\right] < 0$, from which we get $\chi(B) < \chi(B_2)$ (Figures 6b and 6c).

Next, we consider the case where $\chi(B_1) > 0$.

$$f_B\left(\chi(B_2)\right) = f_{B_1}\left(\chi(B_2)\right) + \underbrace{f_{B_2}\left(\chi(B_2)\right)}_{=0}$$
$$= f_{B_1}\left(\chi(B_2)\right) > 0 \ .$$

The last inequality comes from the fact that we have $\chi(B_2) > \chi(B_1)$ and $F_{B_1}$ is increasing on $(\chi(B_1), +\infty)$, so its derivative $f_{B_1}$ is positive on this interval (Figures 6b and 6c). Likewise, we obtain the following equation:

$$f_B\left(\chi(B_1)\right) = \underbrace{f_{B_1}\left(\chi(B_1)\right)}_{=0} + f_{B_2}\left(\chi(B_1)\right)$$
$$= f_{B_2}\left(\chi(B_1)\right) \ .$$

Then, from Lemma 4.3 and $\bar{\lambda}_{B_1} \geq \bar{\lambda}_{B_2}$ (the $\lambda_i$'s are non-increasing), we have the following inequalities:

$$\chi(B_1) = \rho^+(\bar{y}_{B_1}, \bar{\lambda}_{B_1}) \geq \left(\frac{\bar{\lambda}_{B_1}q(1-q)}{2}\right)^{\frac{1}{2-q}} \geq \left(\frac{\bar{\lambda}_{B_2}q(1-q)}{2}\right)^{\frac{1}{2-q}} = m_{B_2} \ ,$$

where $m_{B_2}$ denotes the minimizer of $f_{B_2}$. So, $m_{B_2} \leq \chi(B_1) < \chi(B_2)$. From the variations of $f_{B_2}$ (Figures 6b and 6c), we deduce that $f_{B_2}\left[\chi(B_1)\right] < 0$. To wrap up, we have $f_B\left[\chi(B_1)\right] < 0$, $f_B\left[\chi(B_2)\right] > 0$. Because by assumption $\chi(B_1) < \chi(B_2)$, we get the expected inequality. □

*Proof of Theorem 4.5.* **Necessary conditions** Let $\mathbf{u}$ be a local minimizer of $(P_k)$, *i.e.* $\mathbf{u} \in \mathcal{L}_k$ and let $B = [\![q, r]\!]$ be a maximal block of indices on which $\mathbf{u}$ is constant equal to $\tilde{u}$, i.e. $u_{q-1} > u_q = \cdots = u_r = \tilde{u} > u_{r+1}$. We demonstrate the two points of the theorem:

(i) If $\tilde{u} \notin \{\chi(B), 0\}$, then $\tilde{u}$ is not a local minimizer of $F_B$ (Proposition 4.1). One can thus either infinitesimally increase or decrease $\tilde{u}$ so as to decrease $F_B$ while letting $\mathbf{u}$ feasible. Then, the total objective value $F_{1:k}(\mathbf{u}) = F_{1:q-1}(\mathbf{u}_{1:q-1}) + F_B(\tilde{u}) + F_{r+1:k}(\mathbf{u}_{r+1:k})$ decreases which contradicts $\mathbf{u}$ being a local minima.

(ii) Now, let $\tilde{u} = \chi(B)$ and pick an index $j \in [\![q, r-1]\!]$. We distinguish two cases

- If $\tilde{u} = \chi(B) = 0$, then if $\chi([\![q, j]\!]) = 0$ and/or $\chi([\![j+1, r]\!]) = 0$, either Equation (14) or Equation (15) hold true. It remains to prove that one of these two inequalities is also valid when $\chi([\![q, j]\!]) > 0$ and $\chi([\![j+1, r]\!]) > 0$. In this case we necessarily have $\chi([\![q, j]\!]) \geq \chi([\![j+1, r]\!])$, otherwise Lemma E.1 would contradicts the fact that $\chi(B) = 0$. Hence Equation (15) holds true.

- If $\tilde{u} = \chi(B) > 0$, assume that $F_{q:j}$ is non-increasing at $\tilde{u}$. Then, given that $F_{q:j}$ admits a finite number of critical points minimizers (at most 3), there exists a sufficiently small $\zeta > 0$ such that, for all $\epsilon \in (0, \zeta)$,
$$\tilde{u} + \epsilon < u_{q-1} \quad \text{and} \quad F_{q:j}(\tilde{u} + \epsilon) < F_{q:j}(\tilde{u}),$$
which implies that the point $\boldsymbol{v}$ defined as $\mathbf{u}$ everywhere except on the bloc $[\![q, j]\!]$ where it takes the value $\tilde{u} + \epsilon$ is feasible and verifies $F(\boldsymbol{v}) < F(\mathbf{u})$. This contradicts the local optimality of $\mathbf{u}$. As such $F_{q:j}$ is increasing at $\tilde{u}$. Similarly, we can show that $F_{j+1:r}$ is decreasing at $\tilde{u}$. Then, we get from the possible variations of $F_B$ for a block $B$ (see Figure 6) that
$$\tilde{u} \leq \chi([\![j+1, r]\!]) \quad (\tilde{u} \text{ on a decreasing part of } F_{j+1:r})$$
$$\tilde{u} \leq \chi([\![q, j]\!]) \text{ or } \tilde{u} \geq \chi([\![q, j]\!]) \quad (\tilde{u} \text{ on an increasing part of } F_{q:j})$$

Moreover, when both $\tilde{u} \leq \chi([\![j+1, r]\!])$ and $\tilde{u} \leq \chi([\![q, j]\!])$, we get from $\tilde{u} > 0$ together with Lemma E.1 that $\chi([\![q, j]\!]) \geq \chi([\![j+1, r]\!]) \geq \tilde{u}$, which completes the proof.

**Sufficient conditions** Let $\mathbf{u}$ be such that the conditions of Theorem 4.5 hold. First of all, note that only the last block of $\mathbf{u}$ can be 0. Then, for each block $B = [\![q, r]\!]$ of $\mathbf{u}$ such that its value $\chi(B) > 0$, let define for all $j \in [\![q, r-1]\!]$, $\zeta_{B,j} > 0$ such that $F_{q:j}$ is increasing on $(\chi(B), \chi(B) + \zeta_{B,j})$ and $F_{j+1:r}$ is decreasing on $(\chi(B) - \zeta_{B,j}, \chi(B))$. Note that such $\zeta_{B,j}$ always exists from the conditions of Theorem 4.5.

We define $\bar{\zeta} = \min_{B,j \in B} \zeta_{B,j}$. Now, take $\boldsymbol{v} \in \mathcal{B}(\mathbf{0}, \bar{\zeta})$ such that $\mathbf{u} + \boldsymbol{v}$ is a feasible point. Hence, on each block $B$ of $\mathbf{u}$, the values of $\boldsymbol{v}$ are sorted in decreasing order (to preserve feasibility). Let us look at one block of $\mathbf{u}$, say the first one $B = [\![1, r]\!]$ (wlog), for which we denote by $l \in [\![1, r]\!]$ be the index corresponding to the first negative value of $\boldsymbol{v}$. We have the decomposition $\boldsymbol{v}_B = \sum_{j=1}^{r} \mathbf{w}^j$ where

$$\mathbf{w}^j = (v_j - \max(v_{j+1}, 0))\mathbf{1}_{1:j}, \forall j \in [\![1, l-1]\!] \tag{25}$$
$$\mathbf{w}^j = (v_j - \min(v_{j-1}, 0))\mathbf{1}_{j:r}, \forall j \in [\![l, r]\!] \tag{26}$$

with $\mathbf{1}_{m:n}$ a vector of size $r$ with ones between $m$ and $n$. See Figure 7 for an illustration. Moreover, for all $j \in B$, we have $\mathbf{w}^j \in \mathcal{B}(\mathbf{0}, \bar{\zeta})$. It then follows from the fact that, on $\mathcal{B}(\mathbf{0}, \bar{\zeta})$, $F_{1:j}$ is increasing $\forall j \in [\![1, l-1]\!]$ and $F_{j:r}$ is decreasing $\forall j \in [\![l, r]\!]$ (by definition of $\bar{\zeta}$), that

$$F_B(\mathbf{u}_B) \leq F_B(\mathbf{u}_B + \mathbf{w}^1) \leq F_B(\mathbf{u}_B + \mathbf{w}^1 + \mathbf{w}^2) \leq F_B(\mathbf{u}_B + \sum_{j=1}^{r} \mathbf{w}^j) = F_B(\mathbf{u}_B + \boldsymbol{v}_B).$$

Repeating this for each block of $\mathbf{u}$ and observing that if its last block is 0, then it is on an increasing part of each sub-function over this block, we get that $F(\mathbf{u}) \leq F(\boldsymbol{v})$, for all $\boldsymbol{v} \in \mathcal{B}(\mathbf{0}, \bar{\zeta})$ such that $\mathbf{u} + \boldsymbol{v}$ is feasible. This proves that $\mathbf{u}$ is a local minimizer. $\square$

## F Proof of Theorem 4.6

**Lemma F.1.** *Let $B_1$ and $B_2$ be two successive blocks of indices such that $\max(\chi(B_1), \chi(B_2)) > 0$. Then*

$$\chi(B_1 \cup B_2) \leq \max(\chi(B_1), \chi(B_2)).$$

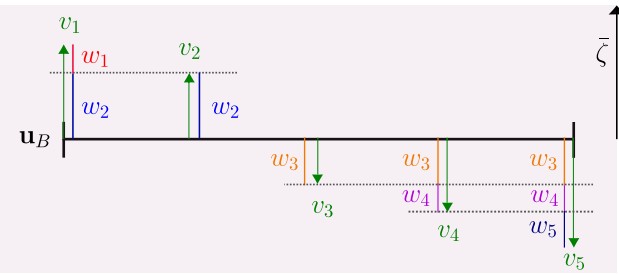

Figure 7: Illustration of the proof of Theorem 4.5.

*Proof.* Let $B = B_1 \cup B_2$ . If $\chi(B) = 0$, then the result trivially holds. If $\chi(B) > 0$, assume that the statement is not true with $\chi(B) > \max(\chi(B_1), \chi(B_2)) > 0$. Hence, from the variations of $F_{B_1}$ and $F_{B_2}$, we get that $\chi(B)$ belongs to an increasing part of each of these two functions. Hence, there exists $\varepsilon > 0$ such that

$$F_B(\chi(B) - \varepsilon) = F_{B_1}(\chi(B) - \varepsilon) + F_{B_2}(\chi(B) - \varepsilon)$$
$$\leq F_{B_1}(\chi(B)) + F_{B_2}(\chi(B)) = F_B(\chi(B)),$$

which contradicts the fact that $\chi(B) > 0$ is the unique positive minimizer of $F_B$. □

*Proof of Theorem 4.6.* Let $\mathbf{x}$ be the solution returned by the PAV algorithm. First of all, by construction of the PAV algorithm, each block of $\mathbf{x}$ satisfies the first point of Theorem 4.5. To prove that the second point of the Theorem is also verified, let $B = [\![q, r]\!]$ be a block of $\mathbf{x}$. If $\chi(B) = 0$, implying it is the last block of $\mathbf{x}$, the second point of Theorem 4.5 is immediately satisfied. From now on, $\chi(B) > 0$. Assume that this block has been involved at most within one backward pass. Then, given the steps of the PAV algorithms, there exists a partition of $B$ in $I$ blocks defined by the indices $q = q_1 < q_2 < \cdots q_I \leq r$ (where $q_i$, is the first index of the i-th block of the partition and we set $q_{I+1} = r + 1$ by convention) such that $\forall i \in [\![1, I]\!]$,

- $\forall j \in [\![q_i + 1, q_{j+1} - 1]\!]$, the singleton $\{j\}$ was added during a forward pass, i.e.,

$$\chi([\![q_i, j - 1]\!]) \leq \chi([\![q_i, j]\!]) \leq \chi(\{j\}) . \tag{27}$$

- The block $\chi([\![q_i, q_{i+1} - 1]\!])$ was merged during a backward pass

$$\chi([\![q_i, q_{i+1} - 1]\!]) \leq \chi([\![q_i, r]\!]) \leq \chi([\![q_{i+1}, r]\!]) . \tag{28}$$

where in both cases the inequalities with the central term are due to Lemma E.1.
Now let take $l \in B \setminus \{r\}$ and denote by $i^*$ the index of the block such that $q_{i^*} \leq l < q_{i^*+1}$. Recall that we want to prove the block $B$ satisfies the second point of Theorem 4.5 via showing Equation (14) *i.e.* $\chi([\![q, l]\!]) < \chi([\![l + 1, r]\!])$. We have from Equation (27) together with the feasibility of the forward pass,

$$\chi([\![q_1, q_2 - 1]\!]) > \cdots > \chi([\![q_{i^*}, q_{i^*+1} - 1]\!]) \underset{(27)}{\geq} \chi([\![q_{i^*}, l]\!]), \tag{29}$$

It then follows from Lemma F.1 that

$$\chi([\![q_1, l]\!]) \leq \chi([\![q_1, q_2 - 1]\!]) \underset{(28)}{\leq} \chi([\![q_1, r]\!]) = \chi(B). \tag{30}$$

Note also that from the previous two equations we have

$$\chi([\![q_{i^*}, l]\!]) \leq \chi(B) . \tag{31}$$

Now, combining the inequalities in Equation (28), we deduce that

$$\chi([\![q_{i^*}, r]\!]) \geq \chi([\![q_1, r]\!]) = \chi(B) . \tag{32}$$

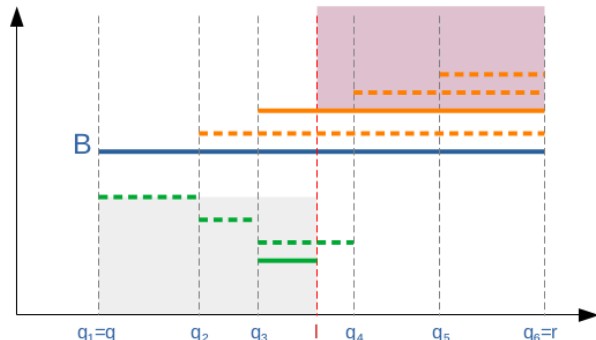

Figure 8: Illustration of the proof of Theorem 4.6. Dashed green blocks illustrate the feasibility of the forward pass while the continuous green block is due to Equation (27). Together, they illustrate the inequalities in Equation (29). Then, inequalities in Equation (30) impose that $\chi(\llbracket q, l \rrbracket)$ lies in the gray area. On the other hand, dashed orange blocs illustrate the constraints in Equation (28) due to the backward pass (with in particular inequalities in Equation (32)). Finally, the red area correspond to Equation (33), highlighting the region were $\chi(\llbracket l, r \rrbracket)$ belongs to. We clearly see that $l$ separate the block in two subblocks satisfying the conditions of Theorem 4.5.

Then, with Equations (31) and (32) and Lemma F.1, we get

$$\chi(\llbracket l, r \rrbracket) \geq \chi(\llbracket q_{i^*}, r \rrbracket) \geq \chi(B) . \tag{33}$$

Finally, with Equations (30) and (33), we conclude that the block $B$ satisfies the second point of Theorem 4.5. If $B$ was the result of multiple backward passes (not at most one as assumed above), we can get the results through the repetition of the above arguments. Finally, repeating that for each bloc of $\mathbf{x}$ shows that the whole $\mathbf{x}$ satisfies the second point of Theorem 4.5. Hence $\mathbf{x}$ is a local minimizer of $F$. $\qquad \square$

# G  Proof for the $\ell_q$ OSCAR model

**Lemma G.1.** *We assume the regularization parameters $(\lambda_i)_i$ follow the OSCAR model. For any successive blocks of indices $B_1$ and $B_2$ such that $\chi(B_1) > \chi(B_2)$, we cannot have $\chi(B_1) > \chi(B_2) > \chi(B_1 \cup B_2)$ with $\chi(B_1 \cup B_2)$ minimizer of $F_{B_1 \cup B_2}$.*

*Proof.* We denote $B \triangleq B_1 \cup B_2$. By contradiction, let assume $\chi(B_1) > \chi(B_2) > \chi(B)$ with $\chi(B)$ minimizer of $F_B$. Then, by Lemma 4.4, we have the following inequality:

$$\chi(B) = \rho^+(\bar{y}_B, \bar{\lambda}_B) \geq \left(\bar{\lambda}_B(1-q)\right)^{\frac{1}{2-q}} .$$

Besides, $f_{B_1}(\chi(B)) > 0$, because $f_{B_1}(\chi(B)) + f_{B_2}(\chi(B)) = 0$ with $f_{B_2}(\chi(B)) < 0$ (because $\chi(B_2) > \chi(B) \geq m_B \geq m_{B_2}$, see Figure 6c). We deduce $\chi(B) < m_{B_1}$ (actually, $\chi(B)$ is smaller than the smallest root of $f_{B_1}$ corresponding to the local maxima of $F_{B_1}$). We obtain the following inequality on $(\lambda_i)_i$.

$$\left(\bar{\lambda}_B(1-q)\right)^{\frac{1}{2-q}} \leq \left(\bar{\lambda}_{B_1}\frac{q(1-q)}{2}\right)^{\frac{1}{2-q}}$$

$$\iff \bar{\lambda}_B \leq \bar{\lambda}_{B_1}\frac{q}{2}$$

$$\iff \frac{(k-m)+(k-n)}{2} = \frac{\sum_{i=m}^{n}(k-i)}{|B|} \leq \frac{q}{2}\frac{\sum_{i=m}^{j}(k-i)}{|B_1|} = \frac{q}{2}\frac{(k-j)+(k-m)}{2}$$

$$\implies k-m \leq 2k-m-n \leq \frac{q}{2}(2k-j-m) \leq q(k-m)$$

which leads to a contradiction as $q < 1$. $\qquad \square$

*Proof of Theorem 4.7.* Let $\mathbf{x}$ be a global minimizer of $(P_k)$ and $i$ be the first index of its last block, i.e. such that $x_i = \cdots = x_k = \tilde{x}$ and $x_{i-1} > \tilde{x}$. We distinguish two cases depending on the value of $\tilde{x}$.

- If $\tilde{x} = 0$, then $\mathbf{x}_{[i-1]}$ must be a global minimizer of $(P_{i-1})$, otherwise this would contradicts the optimality of $\mathbf{x}$ for $(P_k)$. As such, we have $\mathbf{x}_{[i-1]} \in \mathcal{L}_{i-1}$, and $\mathbf{x} \in \mathcal{S}_k$.

- If $\tilde{x} = \chi(\llbracket i, k \rrbracket) > 0$. Then, $\tilde{x}$ is the global minimizer of $F_{i:k}$ (otherwise $(\mathbf{x}_{[i-1]}, 0, \ldots, 0)$ would lead to a better objective value). Let now assume that there exists another local minimizer $(\mathbf{u}^{i^*-1}, \chi(\llbracket i^*, k \rrbracket), \ldots, \chi(\llbracket i^*, k \rrbracket))$ with $i^* > i$. Given that $\mathbf{x}$ is a global (and thus local) minimizer of $(P_k)$, we get from Theorem 4.5, that:

  - either $\chi(\llbracket i, i^* - 1 \rrbracket) \leq \tilde{x} \leq \chi(\llbracket i^*, k \rrbracket)$,
  - $\chi(\llbracket i, i^* - 1 \rrbracket) \geq \chi(\llbracket i^*, k \rrbracket) \geq \tilde{x}$.

  Yet, Lemma G.1 eliminates the case $\chi(\llbracket i, i^* - 1 \rrbracket) \geq \chi(\llbracket i^*, k \rrbracket) \geq \tilde{x}$ (because $\tilde{x}$ is the global minimizer of $F_{i:k}$), so we must have $\chi(\llbracket i, i^* - 1 \rrbracket) \leq \tilde{x} \leq \chi(\llbracket i^*, k \rrbracket)$.
  By construction, $\mathbf{u}^{i^*-1} \in \mathcal{L}_{i^*-1}$ and $u_{i^*-1}^{i^*-1} \geq \arg\min_{x \in \mathbb{R}_+} F_{i^*:k}(x)$.

  - CASE $\arg\min_{x \in \mathbb{R}_+} F_{i^*:k}(x) = \chi(\llbracket i^*, k \rrbracket)$. There exists a block $B$ of $\mathbf{u}^{i^*-1}$ which contains $i$ and we denote it $B = \llbracket s, t \rrbracket$ with $s \leq i \leq t < i^*$. Then by feasibility of $(\mathbf{u}^{i^*-1}, \chi(\llbracket i^*, k \rrbracket), \ldots, \chi(\llbracket i^*, k \rrbracket))$, we have $\chi(\llbracket i^*, k \rrbracket) < \chi(\llbracket s, t \rrbracket)$. Moreover, from the same arguments as used previously, involving Theorem 4.5 and Lemma G.1, we get $\chi(\llbracket i, t \rrbracket) \leq \tilde{x}$. Finally, from the local optimality of $(\mathbf{u}^{i^*-1}, \chi(\llbracket i^*, k \rrbracket), \ldots, \chi(\llbracket i^*, k \rrbracket))$, we obtain from Theorem 4.5 that $\chi(\llbracket i, t \rrbracket) \geq \chi(\llbracket s, t \rrbracket)$. Gathering these different inequalities, we get

    $$\chi(\llbracket i^*, k \rrbracket) < \chi(\llbracket s, t \rrbracket) \leq \chi(\llbracket i, t \rrbracket) \leq \tilde{x} \,,$$

    which leads to a contradiction with $\tilde{x}$ being smaller than $\chi(\llbracket i^*, k \rrbracket)$.
  - CASE $\arg\min_{x \in \mathbb{R}_+} F_{i^*:k}(x) = 0$. It contradicts $\mathbf{x}$ being a global minimizer of $(P_k)$, as one could set $\mathbf{x}_{[i^*:k]}$ to 0 and get a better global objective value.

  To sum up, we cannot have $i < i^*$. Then, if $i > i^*$, it means that $\mathbf{x}_{[1:i-1]} \notin \mathcal{L}_{i-1}$, so one can change some components while remaining feasible and decrease the objective value which contradicts $\mathbf{x}$ being a global minimizer. We finally get that $i$ must equal $i^*$ so $\mathbf{x} \in \mathcal{S}_k$.

$\square$

Proof of Theorem 4.7 focuses on the popular OSCAR sequence, but its conditions can be weakened: as long as $(\lambda_i)_{i \in \llbracket 1, k \rrbracket}$ are chosen such that $\lambda_i = \Lambda(i)$ with $\Lambda$ *concave*, non-decreasing and non-negative, Theorem 4.7 still holds due to the following Lemma G.2.

**Lemma G.2** (Extension of Lemma G.1). *We assume the regularization parameters $(\lambda_i)_i$ write as follows:*

$$\lambda_i = \Lambda(i) \,,$$

*with $\Lambda$ concave, non-increasing and non-negative. Then, Lemma G.1 holds.*

*Proof.* We want to show that for any block $B = [\![m, n]\!]$, any sub-block $B_1 = [\![m, l]\!]$, we have $\bar{\lambda}_B > \bar{\lambda}_{B_1} \frac{q}{2}$.

$$
\begin{aligned}
\bar{\lambda}_B = \frac{1}{|B|} \sum_{i=m}^{n} \Lambda(i) &= \frac{1}{|B|} \sum_{i=m}^{n} \int_{i}^{i+1} \Lambda(i) dt \\
&\geq \frac{1}{|B|} \sum_{i=m}^{n} \int_{i}^{i+1} \Lambda(t) dt && (\Lambda \text{ non-increasing.}) \\
&= \frac{1}{|B|} \int_{m}^{n+1} \Lambda(t) dt \\
&\geq \frac{\Lambda(m) + \Lambda(n+1)}{2} && (\text{Hermite inequality.}) \\
&\geq \frac{\Lambda(m)}{2} && (\Lambda \text{ non-negative.}) \\
&> \frac{q}{2} \Lambda(m) && (q < 1.) \\
&\geq \frac{q}{2} \frac{1}{|B|} \sum_{i=m}^{j} \Lambda(i) && (\Lambda \text{ non-increasing.}) \\
&= \frac{q}{2} \bar{\lambda}_{B_1}
\end{aligned}
$$

$\square$

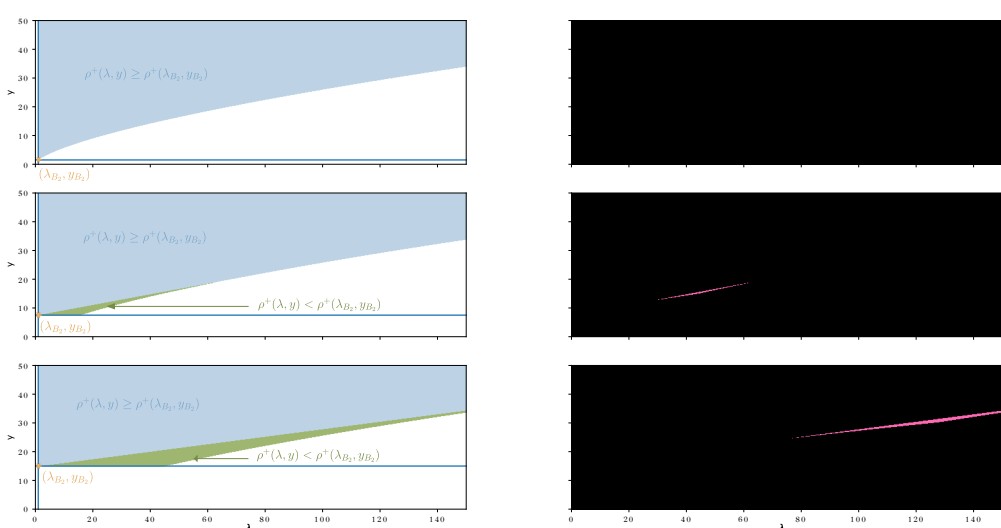

Figure 9: LEFT: In blue, points $(y_B, \lambda_B)$ such that $\chi(B) \geq \chi(B_2)$, in green, $\chi(B) < \chi(B_2)$. RIGHT: Pink points are points $(y_B, \lambda_B)$ such that $\chi(B_1) \geq \chi(B_2) \geq \chi(B)$. Pink points are the only set of points where D-PAV could theoretically fail as it does not consider merging $B_1$ and $B_2$ when $\chi(B_1) \geq \chi(B_2)$.

## H   Details on clustering properties of sorted nonconvex penalties

For every solution $x^*$ of $\min_{\mathbf{x}} f(\mathbf{x}) + \Psi(\mathbf{x})$, there exists a value of $\tau \in \mathbb{R}$ such that $x^*$ also solves

$$
\min f(\mathbf{x}) \quad \text{s.t. } \Psi(\mathbf{x}) \leq \tau. \tag{34}
$$

Hence, solving the the regularized problem can be seen as finding the smallest level line of $f$ that intersect the ball $\{\mathbf{x} : \Psi(\mathbf{x}) \leq \tau\}$, the solution being the corresponding intersection point. To get intuition on how the clustering operates using sorted penalties, Figure 10 displays the projection on the unit ball for various choices of penalties:

- Points of non-differentiability of the penalty attract the solution towards corners of the unit ball. They enforce sparsity for the $\ell_1$ ball and both sparsity and clustering for the SLOPE and sorted $\ell_{1/2}$ unit balls.
- For the $\ell_{1/2}$ unit ball, due to the nonconvexity of the ball, the corners associated with clustering are more or less attractive depending on the choice of the regularization sequence $(\lambda_i)_i$.

We also plot in Figure 11 the level lines of Sorted MCP. We observe that for $\tau$ large enough in the constrained problem, the sublevel set is the whole space (the penalty saturates), the projection is the identity and there is no longer any clustering. Yet, for smaller $\tau$, we still recover the corners associated with clustering.

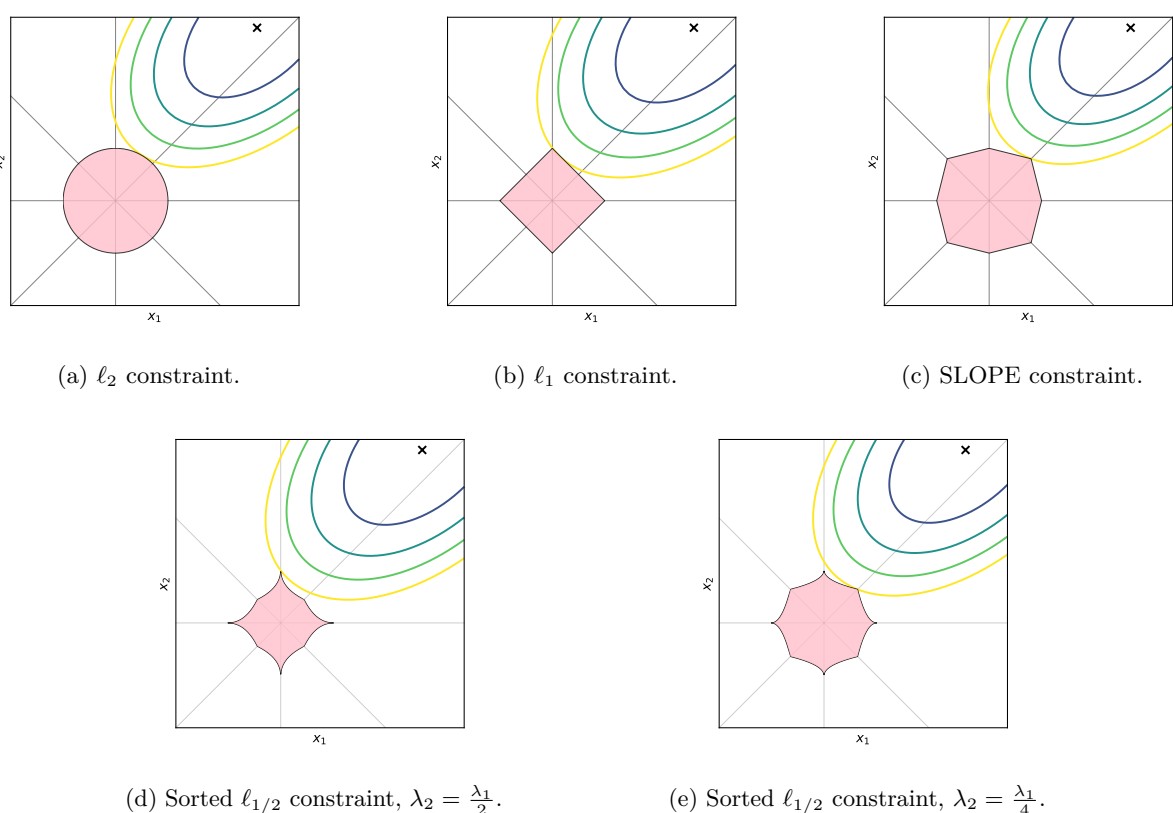

(a) $\ell_2$ constraint.     (b) $\ell_1$ constraint.     (c) SLOPE constraint.

(d) Sorted $\ell_{1/2}$ constraint, $\lambda_2 = \frac{\lambda_1}{2}$.     (e) Sorted $\ell_{1/2}$ constraint, $\lambda_2 = \frac{\lambda_1}{4}$.

Figure 10: Sparse regression in its constrained form. We display the level lines of the least squares datafit and the sublevel set of various penalties (red). *Sparsity*: the non-differentiability of $\ell_1$ attracts the solution towards the corners of the ball. *Structure*: SLOPE also attracts solutions towards points with equal coefficients. These corners are also present in sorted $\ell_{1/2}$ and their attractivity depends on the choice of the regularization sequence.

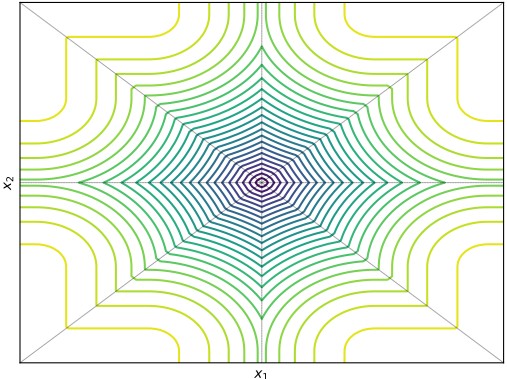

Figure 11: Level lines for the Sorted MCP constraint. $\gamma = 1.4, \lambda_1 = 3, \lambda_2 = 2.5$.

## I  Discussion about the MM approach from Feng & Zhang (2019)

We develop in this section the main differences between our method and the majorization-minimization approach from Feng & Zhang (2019). To that end, we consider the example of Sorted MCP, denoted $\Psi_{\text{SMCP}}$, for which a detailed algorithm is given in Algorithms 2 to 4 of Feng & Zhang (2019). It relies on the decomposition of $\Psi_{\text{SMCP}}$ as a difference of two convex functions. While there is no single such decomposition, given the piecewise quadratic nature of MCP, a natural one is given by

$$\Psi_{\text{SMCP}}(\mathbf{x}, \gamma, (\lambda_i)_i) = \underbrace{\Psi_{\text{SMCP}}(\mathbf{x}, \gamma, (\lambda_i)_i) + \frac{1}{2\gamma}\|\mathbf{x}\|^2}_{\Psi^+(\cdot,\gamma,(\lambda_i)_i) \text{ (convex sorted penalty)}} \underbrace{-\frac{1}{2\gamma}\|\mathbf{x}\|^2}_{\Psi^-}. \tag{35}$$

This allows them to easily deploy a majorization minimization (MM) approach through the majorization of the concave term $-\Psi^-$ by its tangent at the current point. More precisely, the MM iterations are given by

$$\mathbf{x}^{k+1} = \arg\min_{\mathbf{x}} \; f(\mathbf{x}) + (\mathbf{x}^+ - \mathbf{x})^T \nabla \Psi^-(\mathbf{x}^k) + \Psi^+(\mathbf{x}, \gamma, (\lambda_i)_i) \tag{36}$$

where $f$ denotes the datafit, assumed to be differentiable. The minimization of this surrogate function is then tackled with a proximal gradient method. This requires the computation of $\text{prox}_{\Psi^+}$ which can be done using a PAV algorithm. Differently, our approach leverages the weak convexity of the Sorted MCP penalty to use proximal algorithms (such as ISTA, FISTA) directly on the original problem.

We experimentally highlight the difference between these two methods. The setup is the following. The matrix $\mathbf{A}$ is in $\mathbb{R}^{150\times50}$, it is random Gaussian with Toeplitz-structure covariance equal to $(0.4^{|i-j|})_{i,j}$. The dataset is then splitted in 3 equal subsets for training, validatation and test. We denote $n$ the number of samples and $d$ the number of features. The ground truth $\mathbf{x}^\star$ has a sparsity of 60% (i.e. only 40% of non-zero entries) and is made of two clusters of equal coefficients, respectively equal to $+0.5$ and $-0.5$. The penalty used is Sorted MCP with parameter $\gamma$ equal to 1.1 and linearly decreasing regularization strength: $\lambda_i = \alpha \times \frac{d-i}{d}$ for $i \in [\![1, d]\!]$ where $\alpha$ is determined by grid-search.

We consider, using this synthetic dataset, two different settings:

1. Regression: the target $\mathbf{b}$ is defined as $\mathbf{b} = \mathbf{A}\mathbf{x}^\star + \boldsymbol{\epsilon}$ where $\boldsymbol{\epsilon}$ is a vector of white Gaussian noise such that the signal-to-noise ratio is 10. The datafit used is a *least squares datafit* $f(\mathbf{x}) = \frac{1}{2}\|\mathbf{A}\mathbf{x} - \mathbf{b}\|^2$.
2. Classification: The target $\mathbf{b}$ is equal to $\text{sign}(\mathbf{A}\mathbf{x}^\star)$ except for 10% of the samples for which the sign of $b_i$ is flipped. The datafit used is a *logistic regression datafit* $f(\mathbf{x}) = \frac{1}{n}\sum_{i=1}^n \log\left(1 + e^{-b_i(\mathbf{A}\mathbf{x})_i}\right)$.

The algorithms are stopped when the loss difference between two successive iterations is smaller than a tolerance fixed at $10^{-5}$. We consider a MM algorithm with one step of prox-grad at each MM iteration (defined in Equation (36)).

Appendix I display the evolution of the penalized loss value which we aim to minimize:

1. For the least square regression problem, the MM approach from Feng & Zhang (2019) (denoted as LCA SMCP for Linear Convex Approximation Sorted MCP) behaves the same as our method. The convex approximation relies on a quadratic rectification added to the datafit which does not change the behaviour of the loss for the least square case.

2. For the logistic regression problem, the MM approach from Feng & Zhang (2019) has slower convergence rate compared to our direct approach.

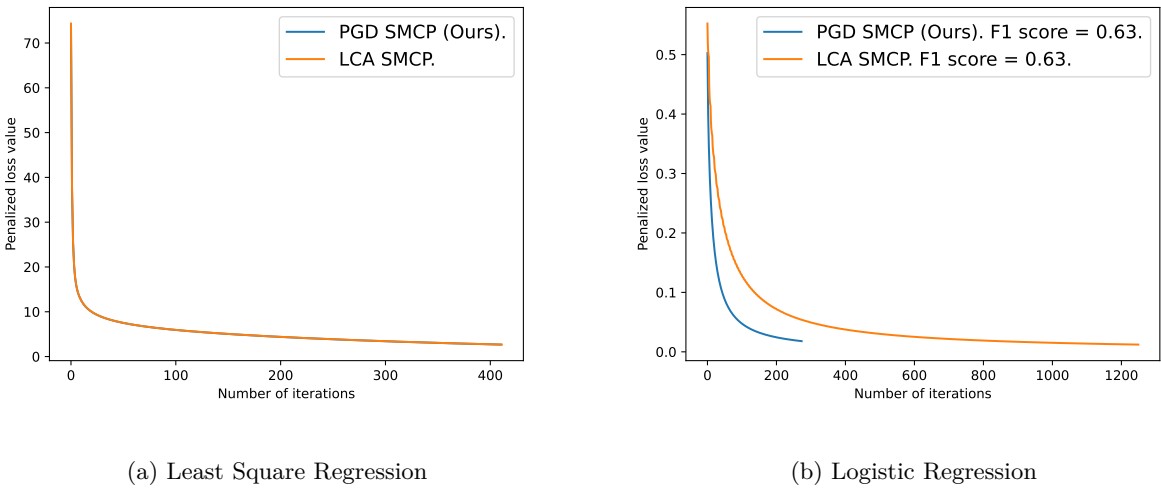

(a) Least Square Regression          (b) Logistic Regression

Figure 12: Comparison of MM and our direct approach on least square and logistic regression problems.

## J    Real data

We illustrate here the solution paths on the *diabetes* data set (Efron et al., 2004). For these numerics, we compare various choices of penalties on a least square regression problem with $n = 442$ and $d = 10$ features. We focus on a low-dimensional problem so as to visualize the solution paths. For the sorted penalties (i.e. SLOPE and Sorted MCP), we take $\lambda = 1.5\lambda^*(1, \sqrt{2} - 1, \ldots, \sqrt{11} - \sqrt{10})$ which corresponds to the quasi-spherical OSCAR sequence (see. Nomura (2020)), where $\lambda^*$ varies along the path. For the MCP and the Sorted MCP penalty, the parameter $\gamma$ is equal to 3. Results are displayed Figure 13. We observe that Sorted MCP indeed clusters features (so does SLOPE when compared to LASSO) and decreases amplitude bias (so does MCP when compared to LASSO). We also observe exact clustering properties when compared to Elastic-Net, which gathers coefficients but does not make them exactly equal.

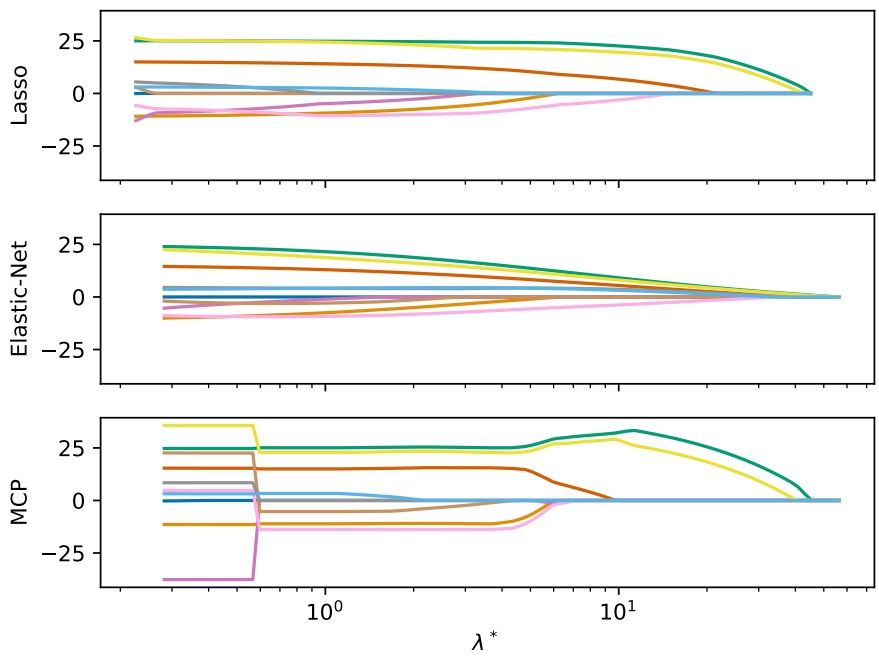

(a) Non sorted penalties

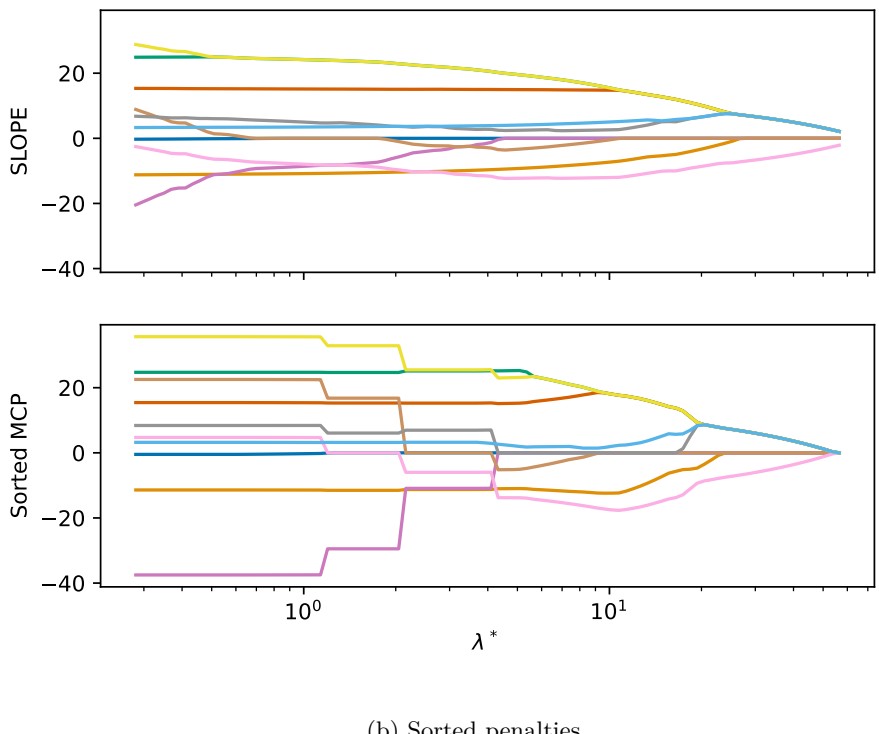

(b) Sorted penalties

Figure 13: Solution paths for convex, nonconvex, sorted and non sorted penalties.

