# OpenReview forum: "Automated and Unbiased Coefficient Clustering with Non Convex SLOPE"
_TMLR — Rejected by TMLR_

### Review · Reviewer_KQjG · 2024-05-09

**Summary Of Contributions:**

The sorted $\ell_1$ norm is defined as follows:
$$
\Psi_{SLOPE}(x)=\sum_{i=1}^{p}\lambda_i|x_{(i)}|,
$$
where $\lambda_1\ge \dots \ge \lambda_p\ge 0$ (with $\lambda_1>0$) are chosen regularization parameters, and $|x_{(1)}|\ge \dots \ge |x_{(p)}|$ represent the sorted components of $x$ in absolute value. This norm is the penalty term of SLOPE, a well-known penalized estimator that promotes sparsity and clustering. The authors introduce alternative penalties to the sorted $\ell_1$ norm by replacing the absolute value in the above expression with a scalar function $\psi$, yielding:
\begin{equation*}
\Psi(x)=\sum_{i=1}^{p}\psi(|x_{(i)}|;\lambda_i).
\end{equation*}
The primary objective of the authors is to compute the proximal mapping of $\Psi$ for various choices of functions $\psi$ in order to derive estimators with improved statistical properties (i.e., ``unbiased'' estimators) compared to SLOPE.

**Audience:**

Yes

**Broader Impact Concerns:**

Not relevant.

**Claims And Evidence:**

Yes

**Requested Changes:**

In addition on important changes described above, some minor changes and typos are listed below:

* item  Page 3,  "we denote by $x_{\downarrow}$ the vector obtained by sorting its components by non-decreasing magnitude,
and $x_{(i)}$ is the i-th component of $x_{\downarrow}$ :
$|x_{(1)} |\ge \dots \ge  |x_{(p)}|$". There is a confusion between $x_{\downarrow}$ and $|x|_{\downarrow}$
and non-decreasing should be replaced by non-increasing.

* item Page 4 $P_{|y|}(y) = |y|_{\downarrow}  \text{ should be replaced by } $

 $P_{|y|}(|y|) = |y|_{\downarrow}$

* item $prob^{-1}$ is the cumulative distribution function of a $N(0,1)$ distribution and not its inverse.

**Strengths And Weaknesses:**

**Strengths:** The authors have adapted the Pool Adjacent Violator (PAV) algorithm to compute the proximal mapping of the function $\Psi$, which appears to be a novel contribution in the literature. The proofs provided are sound.

**Weaknesses:**

**Statistical notions:** While the motivation for this work comes from statistical inference, aimed at estimating the ground truth signal $x^*$ based on the observed signal $x^*+ \text{noise}$, several confusions regarding elementary statistical notions are evident in this article. For instance, in Figure 1 on page 2, the authors write that "nonconvex sorted penalties are less biased (lower RMSE) than the convex SLOPE". This sentence demonstrates a confusion between bias and root-mean-squared error (RMSE). Additionally, the y-axis in this figure represents normalized error rather than the RMSE. Moreover, the expression "sorted penalties are less biased" is abusive; the bias  should refer to estimators, not penalties.

In the section "denoising'' authors construct an "estimator'' $\hat x$, as the image of $x^*+ \text{noise}$  by a proximal operator, namely $\hat x = \text{prox}_{r\Psi}(x^*+ \text{noise})$, where $r>0$ is a scaling parameter. Here are some comments regarding this estimator:

* item $\hat x$ is an unbiased estimator  means  $\mathbb{E}(\hat x)=x^*$, where the expected value is computed with respect to the random vector "noise''. However, it is unrealistic to expect $\hat x$ to be unbiased unless the penalty is null, i.e., if $\hat x=x^*+ \text{noise}$.   Thus, the word "unbiased'' in the title is misleading.

 * item  $\hat x$ is less biased than the SLOPE estimator means
$$ || \mathbb{E}(\hat x)-x^* || \le  || \mathbb{E}(\hat x^{SLOPE})- x^* ||,$$
where $\hat x^{SLOPE}=prox_{r\Psi_{SLOPE}}(x^*+\text{noise})$. However, the authors do not attempt to numerically illustrate or prove this inequality.


I think that the authors are not primarily concerned with the bias; rather, they aim to demonstrate that the estimator $\hat x$ has a lower root-mean-squared error (RMSE), i.e.,
$$\mathbb{E} || \hat x-x^* || \le \mathbb{E} || \hat x^{SLOPE}-x^* ||.$$
However, authors cannot illustrate the above inequality as they  generate only one particular observation of the random vector "noise'' in the numerical experiments.
 To effectively demonstrate numerically improved statistical properties compared to SLOPE, authors should generate at least 1000 (or more) observations of this random vector.
Overall, I strongly recommend that the authors, which are very likely not statisticians, seek assistance from a statistician colleague to ensure the correctness of sections related to statistical comparisons.

**Clustering:** One of the primary advantages of the sorted $\ell_1$ norm is its ability to promote clustering. However, it is unclear why this relevant property holds true when the absolute value in the sorted $\ell_1$ norm is substituted by a scalar function $\psi$. For instance, if $\psi(t)={\bf 1}(t>0)$, then the resulting expression becomes:
$$\Psi(x)=\sum_{i=1}^{p}\psi(|x_{(i)}|;\lambda_i)= \sum_{i=1}^{Card \lbrace j \mid x_j\neq 0 \rbrace }\lambda_i.$$
This expression, depending solely on the support size, does not effectively promote clustering. Therefore, intuitively, sorted log-sum or sorted $\ell_q$ pseudo-norm, which approximate the above expression, may induce degraded clustering properties compared to the sorted $\ell_1$ norm. For example, one could visualize this comment by plotting the unit ball of the penalty $\lambda_1|x_{(1)}|^q+\lambda_2|x_{(2)}|^q$ for various values of $q\le 1$.

---

### Review · Reviewer_YNKz · 2024-06-03

**Summary Of Contributions:**

1. The paper extends the concept of SLOPE (Sorted L1 Penalized Estimation) by introducing a family of nonconvex sorted penalties that enhance feature clustering while reducing the bias associated with large coefficients, common in convex penalties like the traditional Lasso.

2. The authors develop an algorithm that efficiently computes the proximal operator for nonconvex penalties such as sorted MCP (Minimax Concave Penalty) and SCAD (Smoothly Clipped Absolute Deviation). This development includes an extension of the Pool Adjacent Violators (PAV) algorithm.

3. The paper provides a detailed characterization of the solutions to the proximal operator minimization problems associated with nonconvex penalties. This includes identifying conditions under which these solutions are locally and globally optimal, which is crucial for understanding the behavior of optimization algorithms in nonconvex settings.

**Audience:**

Yes

**Claims And Evidence:**

Yes

**Requested Changes:**

1. An explicit and detailed comparison with Feng & Zhang (2019) is necessary. Please delineate step-by-step the differences and similarities between the proposed algorithm and Feng & Zhang’s Algorithm 2, particularly highlighting any innovative enhancements or optimizations that your methodology introduces.

2. The computational cost of the proposed algorithm should be detailed and compared with that of Feng & Zhang (2019), especially since they report a complexity of $O(p \log p)$ for MCP. This analysis will better substantiate claims regarding the efficiency and practical applicability of your algorithm.

3. Please provide the definitions of $\bar{\lambda}_B$ and $\bar{y}_B$ explicitly within Proposition 3.4. Moreover, without these definitions, the proof of Proposition 3.4 is incomplete.

**Strengths And Weaknesses:**

## Strengths

1. The authors extend algorithms to compute proximal operators for a range of nonconvex penalties. These algorithms are not only theoretically sound but also demonstrated to be efficient and effective through rigorous testing.

2. The authors enhance the accessibility and utility of their research by providing an R package implementing the proposed methods.

## Weaknesses

1. The extension of the PAV algorithm to sorted nonconvex penalties such as MCP and SCAD is presented as a novel contribution. However, similar methodologies have already been explored in Feng & Zhang (2019), where a PAV-based approach was proposed for computing the proximal operator of sorted nonconvex penalties including MCP. The details can be referred to Algorithm 2 in Feng & Zhang (2019). This significant overlap raises questions about the novelty of the current paper's contributions unless further differentiation is clearly articulated.

2. The assertion that this work differs from Feng & Zhang (2019) by avoiding the use of a majorization step in computing the proximal operator appears to be incorrect. Feng & Zhang (2019) indeed utilize a majorization-minimization approach but do not apply majorization in the proximal mapping computation itself. This misstatement could mislead readers regarding the true novelty and technical contribution of the current paper.

---

> ### Author Response · Authors · 2024-06-14
>
> We thank you for your comments, which help us to clarify our contribution compared to Feng & Zhang.
> We have enriched the discussion by adding some experiments in Annex I, directly comparing our method and the one from Feng & Zhang. We have also recalled the notation $\bar y_B$ in proposition 3.4.
> We provide below some additional comments on the novelty of our approach.
>
> ### Contributions
>
> First, we would like to highlight that our work aims to propose a **unified framework** on the practical use of sorted penalties, not restricting ourselves to the case of SLOPE.
> To this extent, we tackle MCP and SCAD which was previously done by Feng & Zhang in a slightly different manner but we also handle a wider family of non convex penalties (log sum, $\ell q$ for $0 < q <1$), providing explicit algorithm and code to compute their proximal operator.
>
> ### Discussion on Feng & Zhang
>
> The main difference between our approach and the method proposed by Feng & Zhang is that we explicitly rely on the weak convexity of MCP and SCAD to make  use of proximal algorithms (without majorization-minimization approach).
> We have detailed this in Appendix I.
> - In Algorithm 2 of Feng & Zhang, the prox is not computed on MCP directly but on $\rho_+$ which is the sum of MCP and some quadratic regularization (thus, the majorization step).
> We argue our approach is more direct as our computation enables the use of proximal algorithms with guarantees of convergence using weakly convex penalties (Douglas-Rachford, Forward Backward).
> - Yet, we agree with the reviewer that our prox computation is similar to the one proposed by Feng & Zhang as they both rely on PAV algorithm which is frequently involved in such sorted problems.
> - For least-square regression problem, the quadratic rectification used by Feng & Zhang does not change the behavior of the algorithm as the datafit is itself quadratic.
> Nevertheless, for logistic regression, it is no longer the case and we observe slower convergence rate when using their MM approach compared to ours (see Annex I added to the document).
>
> ### Regarding complexity
>
> We have the same complexity as their algorithm for computing the proximity operator.
> Indeed, both our algorithms rely on PAVA which has linear complexity once the input vector is sorted (hence, quasi linear complexity) as long as the pooling operation is made in $\mathcal O(1)$.
> What differs is that we calculate the prox of Sorted MCP while Feng & Zhang calculate the prox of the sum of Sorted MCP and a  Quadratic.

---

> > ### Author Response · Authors · 2024-06-28
> >
> > I hope our rebuttal has answered your questions. If there are still some points which need to be clarified, we will be pleased to provide additional comments.

---

### Review · Reviewer_16P9 · 2024-06-07

**Summary Of Contributions:**

The paper investigates the optimization aspects of sorted non-smooth regularizers used for sparse structured generalized linear models. Considering the framework of proximal methods, the authors show how prox for the mentioned regularizers can be computed. These regularizers are then generalized to non-convex case and studied from the perspective of local and global minimizes of the corresponding proximal operator optimization problem. Finally, the paper concludes with illustrative synthetic experiments that validate the theory and show the advantages of considered regularizers.

**Audience:**

Yes

**Claims And Evidence:**

Yes

**Requested Changes:**

Adding a discussion on practical motivation and real-world applications.

Fixing issues mentioned in the Strengths And Weaknesses part.

**Strengths And Weaknesses:**

## Strengths
- The manuscript is well-organized, and the writing is detailed and mostly clear.
- All statements are accurately presented with links to the corresponding proofs in the Appendix.
- A good literature overview describes the prior works, their key features, and disadvantages.
- Experiments are carefully designed to illustrate the key advantages of the studied techniques.

## Weaknesses
- The paper lacks practical motivation and application examples. The introduction clarifies the benefits of the proposed penalties, but it is not apparent whether this is relevant for real-world scenarios. Experiments are also performed only for synthetic problems.
- Most references are (stats/opt) journal papers, while few are Machine Learning (ML) venues. This makes me question the suitability of TMLR for this work, especially given that I was assigned to review this paper while not being an expert on the topic. I would suggest adding more detail to put the paper into a broader ML context to expand the audience.
- In contributions, the authors say
> we numerically show that the proposed PAV-like algorithm does not reach the global solution only for very specific cases.

While this is appreciated, rigorous guarantees are not provided. Thus, it has not been proven that the PAV-like algorithm can not reach the global solution. Maybe for some other adversarial situations, it is indeed the case.

- Starting from Subsection 4.2, the notation becomes quite heavy and unclear. Overall, I think the main text is overloaded with technical details. It may be beneficial for a broader audience maybe to move some of the details to the appendix and add more comments and explanations, e.g., on what the advances are in terms of (proof) techniques and novelty in the analysis.

### Questions
__Q1.__ Why are these particular numbers/values used in the Experiments (e.g., in Subsection 5.1)?

__Q2.__ Legend of Figure 2 mentions
> optimal regularization strength
What does “optimal” mean here?


#### __Minor__
I would suggest replacing Definition 3.1 and Assumption 1 with just one assumption that combines it with the definition.

Typo: missed word in Remark 3.3
> note that it consistent with

Typo: lacks summons at the very end of Example 3.6

Numerical characteristics (performance metrics) like those in Figure 1 would be appreciated for Figure 3.

The paper ends somewhat abruptly. The conclusions section would be helpful.

There are issues with some of the references, such as a lack of journal names or links. E.g., Feng and Zhang, 2019; Su & Candès, 2018

---

> ### Author Response · Authors · 2024-06-14
>
> We thank you for your comment and your remarks. We have fixed typos and incomplete citations. We have added a conclusion as you suggested as well as some additional experiments detailed below.
>
>
> ### On research scope
>
> We believe our work falls within the scope of machine learning research, as our approach enriches the methods on regularized generalized linear models.
> We provide below some examples of papers which propose proximal algorithms for non-convex regularized problems, published in ML venues.
> 1. Gong, P., Zhang, C., Lu, Z., Huang, J., & Ye, J. A general iterative shrinkage and thresholding algorithm for non-convex regularized optimization problems. ICML (2013): the authors propose a proximal gradient algorithms with proofs of convergence for non-convex regularized problems (encompassing SCAD, MCP, Capped $\ell_1$ but not $\ell_q$ with $0 < q < 1$).
> 2. Yao, Q., & Kwok, J. T.. Efficient learning with a family of nonconvex regularizers by redistributing nonconvexity. In JMLR (2018): the authors propose a general framework to tackle problems with non-convex penalties such as MCP and SCAD relying on DC decompositions.
> 3. Rakotomamonjy, A., Gasso, G., & Salmon, J. Screening rules for lasso with non-convex sparse regularizers. In ICML (2019): the authors propose MM algorithms with screening rules for regression problems with (non-sorted) MCP or SCAD penalties.
>
> There are many other similar works published at ICML, Neurips or JMLR, and we can provide more references if needed.
>
>
> ### On practical aspects
> We added some context in the introduction to underline how these models are used in practice (applications to compressed sensing, to portfolio selection, to neural network compression).
>
> - The main advantage of sorting penalties is that it promotes solutions with inner structure (i.e. clusters of equal coefficients) which enables one to identify groups of correlated features.
> It has interests in all regression problems where one wants insights on the structure of the solution (for explainability purposes for instance).
> - The main advantage of non-convexity is to recover solutions with less amplitude bias, which experimentally leads to solutions with smaller estimation error.
>
> We have also added in Appendix J some experiments on real data, computing solution paths for various choices of penalties: $\ell_1$, Sorted $\ell_1$, MCP, Sorted MCP, etc.
>
>
>
> ### On global convergence and counter-examples
>
> We have changed the sentence *"we numerically show that the proposed PAV-like algorithm does not reach the global solution only for very specific cases."* to *"Experiments tend to indicate that the proposed PAV-like algorithm may not reach the global solution only for very specific settings and we were not able to build such counter-examples."*
>
>
> ### Section 4.2
>
> In section 4, the content of the proofs are in appendix. We focussed on giving insight through proof sketches and high level explanations. To our knowledge, this section on the sorted $\ell_q$ is quite novel in the way we handle this isotonic nonconvex optimization problem, hence some degree of technicality is unavoidable. Section 5.3 also gives more explanations on how our algorithms works and its limitations.
>
>
> ### Answers to questions
>
> *Q1. Why are these particular numbers/values used in the Experiments (e.g., in Subsection 5.1)?*
>
> Several choices have been made:
> - the choice of the values of the ground truth vector is somewhat arbitrary. We have chosen such clusters for visualization purposes (to clearly identify the different clusters of the solution).
> - the choice of the regularization sequence is made on the OSCAR sequence (which is used a lot with SLOPE), i.e. a sequence of linearly decreasing regularization parameters $(\lambda_i)_i$.
> -  For regression (section 5.2), the choice of the matrix $\mathbf A$ is made to enforce strong correlation between features as it is the setting where sorted penalties are known to outperform standard penalties.
>
> Besides these remarks, the code is attached to the paper and will be made public so as to enable anyone to carry experiments on modified settings.
>
> *Q2. optimal regularization strength What does “optimal” mean here?*
>
> The definition was actually missing, we thank you for pointing it out.
> The dashed line represents the regularization strength for which we have a good tradeoff between clustering and estimation error.
> It has been clarified in the paper.

---

### Comment · Action_Editor_j3eh · 2024-05-23
**Sorry for the delay**

Dear authors,
  As you can see, one review is in, and hopefully the 2nd review will be in soon.  The 3rd reviewer had a medical emergency this month and isn't able to finish the review, so I will look for a replacement.  Sorry for the delay.

Best,
Stephen

---

### Author Response · Authors · 2024-06-14

We thank all 3 reviewers for their careful feedback.
All changes made to the paper are written in **blue**.
We would like to point out the three main additions made in light of the reviews:
- Appendix section H provides more insight on clustering properties of sorted penalties.
- Appendix section I highlights the differences between Feng and Zhang approach (2019) and ours.
- Appendix section J provides experiments on real data.

---

### Comment · Action_Editor_j3eh · 2024-06-21
**Discussion phase**

Dear authors and reviewers,

We're in the discussion phase. The authors have provided rebuttals to the initial comments from the reviewers, and now is the time for reviewers to look at these rebuttals and respond (and in turn, authors can respond to these new comments). In my opinion, this is the best part of TMLR: we can have a dialog, rather than the old-fashioned method of turning in comments and waiting a month for a written response.

Best,
Stephen (action editor)

---

> ### Comment · Action_Editor_j3eh · 2024-07-06
> **Final phase**
>
> Dear authors and reviewers,
>
> We're starting the final phase. Reviewers, please submit an "Official Recommendation" asap (and please include justification! I won't just blindly average the numerical scores). These recommendations are private to me and the EiC, and then we'll use them to make a public decision.
>
> Authors, if you have any final comments, you're still welcome to post (or if that's no longer enabled, you can post to me as the reader and I can repost so the reviewers can see).
>
> -Stephen

---

### Comment · Reviewer_KQjG · 2024-06-27
**Amplitude bias**

In the updated version of the article, the authors introduce the notion of amplitude bias. This is not a concept that I use
(to compare estimators, I only use the notions of bias and mean squared error). What is the definition of the amplitude bias? For instance, the amplitude bias of the noisy signal "x* + noise" is very likely explicit; what is its expression?

---

> ### Author Response · Authors · 2024-07-09
>
> Thank you for your feedback on the updated version of the document, we have added explanation in the paper for the term "amplitude bias".
> It refers to the shrinkage of non-zero coefficients toward $0$ due to the $\ell_1$ penalty term in the Lasso, i.e. the fact that the amplitude of non-zero coefficients tends to be underestimated by a factor depending on the regularization strength ($\lambda$ when using $\lambda \lVert \cdot \rVert_1$).
> Below we give a few more references from the ML literature that use "bias" or "amplitude bias" as we do, and we also provide an explicit calculation for the $x^\star+noise$ case you pointed out.
>
> It is often referred to directly as "bias" in lots of ML venues, see for example:
> - [1] " For instance (Fan & Li,2001) proposed the Smoothly Clipped Absolute Deviation (SCAD) penalty to circumvent the drawbacks of the Lasso penalty, in particular with respect to bias."
> - [2] "A possible drawback to the contextual lasso, and indeed all lasso estimators, is bias of the linear model coefficients towards zero. This bias, which is a consequence of shrinkage from the l1-norm, can help or hinder depending on the data."
>
> Your first review encouraged us to be more explicit by adding the term "amplitude", the full expression "amplitude bias" can also be found in the literature, we provide below some quotes:
> - [3]: The authors compare group convex penalties (group-Lasso) and group non-convex penalties.
> Their use of amplitude bias refers to the same phenomenon as the one we highlight in our paper: non-convex penalties mitigate the shrinkage of non-zero coefficients which is observed with Lasso.
> "The $\ell{2,1}$ norm recovers the four sources with an amplitude bias (the estimated amplitude is lower than the exact one). [...] The $\ell{2,0.5}$ norm estimates the exact four source amplitudes with- out amplitude bias thanks to the non-convexity."
>
> - [4]: "when using convex sparsity promoting priors, one needs to compensate for the amplitude bias due to the shrinkage of the weights".
>
> [1] Lozano, A. C., & Swirszcz, G. (2012, June). Multi-level lasso for sparse multi-task regression. In Proceedings of the 29th International Coference on International Conference on Machine Learning (pp. 595-602).
>
> [2] Thompson, R., & Dezfouli, A. (2024). The contextual lasso: Sparse linear models via deep neural networks. Advances in Neural Information Processing Systems, 36.
>
> [3] Bekhti, Y., Badeau, R., & Gramfort, A. (2017, August). Hyperparameter estimation in maximum a posteriori regression using group sparsity with an application to brain imaging. In 2017 25th European Signal Processing Conference (EUSIPCO) (pp. 246-250). IEEE.
>
> [4] Gramfort, A., Thirion, B., & Varoquaux, G. (2013, June). Identifying predictive regions from fMRI with TV-L1 prior. In 2013 International Workshop on Pattern Recognition in Neuroimaging (pp. 17-20). IEEE.
>
>
> ### Computation of the amplitude bias
>
> Let us illustrate this amplitude bias with the pointed example $y = x^* + n$ where $n \sim \mathcal{N}(0,\sigma^2)$. The associated LASSO estimator is the well known soft thresholding $\hat x_\lambda = \mathrm{sign}(x)  \mathrm{max}(|x| - \lambda,0)$ which can be rewritten as $\hat x_\lambda = \hat x^1_\lambda +\hat x^2_\lambda$ with $ \hat x^1_\lambda = \mathrm{max}(x - \lambda,0)$ and $ \hat x^2_\lambda = \mathrm{min}(x + \lambda,0)$.
> The expectation $\mathbb{E}(\hat x^1_\lambda)$ (similarly for $\mathbb{E}(\hat x^2_\lambda)$) can be computed in explicit form through the decomposition of the corresponding integral in the sum of two integrals with appropriate bounds. Few calculations lead to
> $$\mathbb{E}(\hat x^1_\lambda) = \sigma \phi\left(\frac{\lambda -x}{\sigma}\right) + (x - \lambda) \left(1 - \Phi \left(\frac{\lambda - x}{\sigma} \right) \right)$$
> $$\mathbb{E}(\hat x^2_\lambda) = - \sigma \phi\left(\frac{\lambda -x}{\sigma}\right) + (x - \lambda) \Phi \left(\frac{\lambda - x}{\sigma}  \right)$$
> where $\phi$ denotes the standard normal PDF and $\Phi$ the standard normal CDF. This gives us a theoretical expression of the bias: $\mathbb{E}(\hat{x}_\lambda) - x^*$.
>
> We uploaded in the supplementary material plots where we see that the bias does correspond to $\pm \lambda$ when $|x^*|$  is large (the amplitude bias in sparse recovery literature) and is linear around $x^* = 0$.

---

> > ### Comment · Reviewer_KQjG · 2024-07-21
> > **Response to authors' comments**
> >
> > Thank you for the clarification.
> >
> > Regarding articles [3] and [4], which are recent and very applied, the term "amplitude" seems to have a specific meaning related to applications. This term is used in phrases like "amplitude source" or "amplitude voxel." In this article, I do not believe that renaming a well-known notion like "bias," which was introduced nearly a hundred years ago, helps to clarify this article.
> >
> > Since the authors renamed "bias" to "amplitude bias" after the first round of the revision, I expected "amplitude bias" to denote a different concept from the traditional bias. Given that "bias" and "amplitude bias" represent the same notion, my question was naive because the noisy signal x* + noise is both the maximum likelihood estimator for x* and the best unbiased estimator; hence, its bias is null. Actually, I did not understand why the authors attempted to derive the bias of the LASSO (moreover there are numerous errors in their calculations).
> >
> > Furthermore, I still do not understand how the authors can claim in Figure 1 that "nonconvex sorted penalties mitigate the amplitude bias compared to convex SLOPE" when this experiment was conducted with only one realization of the noise. This claim is particularly surprising given that, in deriving the bias for the LASSO, the authors computed an expected value. Authors probably know that an expected value cannot be accurately approximated with a single observation.
> >
> > Regarding the comment: "the convex L1 norm (the Lasso, Tibshirani, 1996) has the drawback of shrinking all nonzero coefficients towards 0. In the sparse recovery literature, this undesirable property is commonly called amplitude bias. It can be mitigated by using nonconvex penalties", I would like to point out that, for almost 70 years, it has been known that shrinking components toward zero provides a biased estimator that may have a smaller mean squared error than the noisy signal (see, for example, the Wikipedia page on the James-Stein estimator). If the authors really want an unbiased estimator, the best approach is not to use penalization; after all, the noisy signal is unbiased.
> >
> > Overall, in the current version no one numerical experiments in this article corroborate the following sentences:
> >     • In the introduction: "...but are less biased than their popular convex counterpart,"
> >     • In the conclusion: "... but promote solutions with less amplitude bias."

---

> > > ### Author Response · Authors · 2024-07-22
> > >
> > > We thank you for your help on framing our paper, and chosing the appropriate terms.
> > >
> > > First of all, we would like to recall that our contribution is to propose an efficient and practical solution for sorted non-convex penalties in practice through the computation of their proximal operator. Our contribution is not a statistical one.
> > >
> > > - Non-convex penalties in their nonsorted form are widely used in sparse optimisation.
> > > This is supported by arguments that they lead to estimates that are less biased than the L1 penalty (see the original paper of SCAD and MCP mentioned in our first answer).
> > > As we have mentioned in the introduction, one may *expect* non-convex sorted penalties to exhibit the same interesting behavior.
> > > Clearly, we have not focused on studying the statistical properties of the estimator: our approach is centered on optimization with the derivation of their proximal operator.
> > > Perhaps the title is misleading and suggests that a theoretical study of bias has been conducted. We propose to rename it "Proximal Operators of Non Convex SLOPE".
> > >
> > > - Regarding the *empirical* quantification of bias, we point the reviewer to Figure 2, of which our Figure 1 is a particular example showing *on one sample only* how each penalty acts (while figure 2 is a repetition over 1000 samples, and a large range of regularization strengths).
> > > We expect the better amplitude recovery by sorted non-convex penalties to be a consequence of unbiasedness (as it has been observed in the non-sorted case).
> > > Figure 1 is displayed at the start of the paper for illustrative purposes, i.e. to get a quick understanding of the properties of sorted non-convex penalties.
> > > We have clarified in the paper that Figure 1 is of course not in itself a proof of unbiasedness, and we refer to Figure 2 which does compute the F1 score and relative MSE on 1000 samples of the noise.

---

### Decision · Action_Editor_j3eh · 2024-07-25

**Recommendation:** Reject

**Comment:**

While one reviewer was positive, two other reviewers were ultimately firmly negative, and at least one of them (KQjG) engaged in a few rounds of discussion.

There were three main criticisms:
- from an optimization point of view, reviewers found that this paper was quite similar in results to Feng and Zhang. The authors point out that their approach is different and more general than the Feng and Zhang paper, which the reviewers seem to believe, but most reviewers just found the difference and novelty to be rather slight, and/or were not convinced that the added generality was useful for practical problems.
- one reviewer initially wrote that the methods lacked practical motivation. This was partly addressed by the rebuttal, but overall, reviews would have been more positive if there was a very clear practical example that showed more benefit.  For example, section 5.2 had examples on regression, but this was synthetic (maybe designed more to illustrate technical results rather than to illustrate utility).
- another reviewer had reservations about the statistical terminology (and some derivations). This reviewer's basic point was that for denoising an additive noise problem ($y=x^\star + \epsilon$) for noise $\epsilon$, asking for a good unbiased estimator makes little sense because the Gauss-Markov theorem implies that $\hat{x}=y$ is the best (in terms of MSE) unbiased estimator. The authors seem to mean just that non-convex penalties are often **less** biased than convex penalties (and they offered to change the title of the paper to remove the "unbiased" word), but overall this section could be clarified and the numerical experiments could be adjusted to more clearly demonstrate exactly what the authors mean.

Each of these criticisms by itself is not strong enough to reject the paper, but taken together I unfortunately think the paper would be better if it had a major revision and was either resubmitted to TMLR or submitted elsewhere (criticisms 2 and 3 are certainly addressable). In particular, given that the optimization contribution exists but is slight, the practical/statistical motivation needs to compensate, and the reviewers and I did not find it adequately did so in the current manuscript.

Regarding the TMLR criteria:
- `Are the claims made in the submission supported by accurate, convincing and clear evidence?`
  - One reviewer found the discussion about bias confusing and not supported by the experiments
- `Would some individuals in TMLR's audience be interested in the findings of this paper?`
  - The similarity to Feng and Zhang, and the lack of concrete example showing the benefit of the method on a problem (even a stylized problem) limit the immediate interest to the community

**Audience:**

These proximity operators arise as subproblems in sparse recovery, clustering, and some neural net compression applications.  One reviewer initially raised concerns that this was targeted toward statisticians rather than machine learning researchers, though this reviewer was mostly assuaged by the authors' response (they noted other similar papers in ML venues).  I agree with the authors that this paper has sufficient ML content to interest at least a sizable fraction of TMLR readers.

**Claims And Evidence:**

The authors derive the proximity operators of a class of nonconvex sorted functions, generalizing the sorted L1 norm (of SLOPE) and also generalizing more recent work by Feng and Zhang.

The reviewers found no issues with the optimization (proximity computation) derivations and correctness.

The statistical / signal-processing benefit was based on numerical experiments. Not all reviewers found these convincing (in particular, reviewer KQjG didn't find them to sufficiently demonstrate the unbiasedness, even after the revisions).

**Resubmission Of Major Revision:**

The authors may consider submitting a major revision at a later time.